

# Estimating the refractivity bias of Formosat-7/COSMIC-II GNSS Radio Occultation in the planetary boundary layer

Gia Huan Pham[1], Shu-Chih Yang[1,2], Chih-Chien Chang[1], Shu-Ya Chen[2], and Chung-Yung Huang[3]

[1]Department of Atmospheric Sciences, National Central University, Taoyuan, Taiwan
[2]GPS Research and Application Center, National Central University, Taoyuan, Taiwan
[3]Taiwan Space Agency, Hsinchu, Taiwan

*Correspondence to*: Shu-Chih Yang (shuchih.yang@atm.ncu.edu.tw)

**Abstract**

FORMOSAT-7/COSMIC-2 radio occultation (RO) measurements are promising for observing the deep troposphere and providing critical information on the Earth's planetary boundary layer (PBL). However, refractivity retrieved in the low troposphere can have severe bias under certain thermodynamic conditions. This research examines the characteristics of bias in the low troposphere and presents methods for estimating the region-dependent bias using regression models. The results show that the bias has characteristics that vary with land and oceans. With substantial correlation between local spectral width (LSW) and bias, the LSW-based bias estimation model can explain the general pattern of the refractivity bias but with deficiencies in measuring the bias in the ducting regions and certain areas over land. The estimation model involving the relationship with temperature and specific humidity can capture the bias of large amplitude associated with ducting. Finally, a minimum variance estimation that combines the benefits of the individual estimation provides the most accurate estimation of the refractivity bias.

**1 Introduction**

Global Navigation Satellite System (GNSS) radio occultation (RO) observations have become a critical data source in atmospheric applications, particularly numerical weather prediction (NWP) (e.g., Healy, 2008; Rennie, 2010; Cucurull et al., 2007, 2017; Lien et al., 2021). Low-Earth-orbiting (LEO) satellites receive radio signals, which are emitted from GNSS transmitters and tend to bend due to atmospheric density changes. Information on the bending angle can be obtained with the GNSS RO technique, and then the atmospheric refractivity is further derived by Abel inversion. Since the RO technique measures the signal phase delay, it is not affected by clouds and rainfall. The RO profile is an all-weather observation with a high vertical resolution.

The RO observations, bending angle and reflectivity, reflect the changes in atmospheric density, a function of temperature, moisture and pressure (Kuo et al., 2004). RO observations were indicated to be advantageous in providing information on temperature (stratosphere and upper troposphere) and moisture (lower troposphere) with low noise and low systematic errors, which is very beneficial in atmospheric research (Eyre, 2008). Several GNSS RO missions, e.g., the FORMOSAT-3/Constellation Observing System for Meteorology, Ionosphere, and Climate (FS3/C), FORMOSAT-7/COSMIC-2 (FS7/C2), Meteorological Operational satellite (MetOp), Gravity Recovery And Climate Experiment (GRACE), Satellite de Aplicaciones Cientifico-C (SAC-C), X-band TerraSAR satellite (TerraSAR-X), Korea Multi-Purpose Satellite-5 (KOMPSAT-5), etc., have provided much RO data for numerical



weather prediction (NWP). Many works have illustrated the positive impact of assimilating RO observations, such
as the operations systems at the European Centre for Medium-Range Weather Forecasts (ECMWF) (Healy, 2014),
the NCEP/Environmental Modeling Center (EMC) (Cucurull, 2007) and the Taiwan Central Weather Bureau
(CWB) (Lien et al., 2021). Moreover, studies have been initiated recently to investigate the potential of
assimilating the large volume of commercial RO data from Spire, and the benefits can be identified in weather
forecasting (Bowler, 2020a). In addition to improving global NWP, studies have also confirmed that assimilating
RO observations improves severe weather prediction, particularly for tropical cyclones and heavy rainfall (e.g.,
Chen et al. 2020; 2021a,b, 2022; Chang and Yang, 2022; Yang et al., 2014).

46        As the successor of FS3/C, the FS7/C2 mission was launched in 2019 with support from the Taiwan
National Space Agency (TASA) and the United States National Oceanic and Atmospheric Administration
(NOAA). The number of profiles obtained by FS7/C2 is approximately three times greater than that of FS3/C
since FS7/C has dense coverage over the tropics and subtropics (Chen et al., 2021c). Compared with FS3/C,
FS7/C2 has a higher signal-to-noise ratio (SNR), wider bandwidth, and a better open-loop (OL) model. These
advantages enable the retrieval of more data from RO signals penetrating the moist troposphere and having the
ability to detect the atmospheric boundary layer (ABL) and superrefraction (SR) over the top of the planetary
boundary layer (PBL) (Schreiner et al., 2020). Chen et al. (2021c) showed that the data availability of the FS7/C2
RO profiles under 1km is five times greater than that of the FS3/C profiles over a six-month range. Anthes et al.
(2022) noted that the penetration rate of RO profiles is limited to extremely moist conditions and that the rate is
high near tropical cyclones and their environment. It is expected that FS7/C2 will continue to experience the same
success as its predecessor (FS3/C) given the quality and quantity of data collection with advanced improvement
in measuring techniques (Feng et al., 2020). The ability to penetrate deep into the atmosphere makes RO
measurements ideal for studying the PBL. The PBL is directly influenced by any exchange of energy, momentum
and mass between the Earth's surface and the atmosphere, and thus its characteristics are crucial for weather and
climate variabilities.

62        However, the use of GNSS RO in the lower atmosphere still has uncertainties when radio rays pass through
areas with strong refractivity gradients. In such conditions, the assumptions and approximations in the retrieval
algorithms can result in large uncertainties in the RO data (Sokolovskiy, 2010). Normally, when the refractivity
gradient is small, the radio rays can converge with a given impact parameter well, and the wave optics
transformation (WO) technique can retrieve complicated RO signals efficiently (Gorbunov, 2002; Jensen et al.,
2003, 2004). However, in the presence of a strong vertical refractive gradient, multipath propagation can extend
the spectrum of WO-transformed RO signals, resulting in complex structures in the RO bending angle
(Sokolovskiy et al., 2010), hence causing the complexity of RO uncertainty estimation. In this case, the systematic
error induced by the tropospheric strong refractivity causes a negative refractivity bias ($N$-REFB) (Rocken et al.,
1997). The $N$-REFBs in the lower troposphere are largely attributed to the existence of the ducting layer (Xie et
al., 2010), which results in significant changes in both the phase and SNR of the RO signals (Sokolovskiy, 2003)
and thus leads to bending angle errors and additional refractivity errors. Ao (2007) demonstrated that the GPS RO
$N$-REFB has latitudinal and monthly variations below the 2-km height. The climatological locations of the $N$-
REFB agree well with the areas of high ducting frequency, mainly over the subtropical eastern oceans (Feng et
al., 2020). However, in addition to ducting, issues such as tracking error, cycle slips and unbalanced noise spectra



could also lead to lower-altitude *N*-REFBs. In regard to the assimilation of RO data, quality control (QC) is applied
to reject the RO data if the observation or the corresponding backgrounds are suspected to be affected by super
refraction. The rejection rate is high below 2 km due to the negative bias, which could also discard valuable
information for data assimilation. To increase the value of RO data in the lower atmosphere, this study aims to
examine the characteristics of the REFBs in more detail and proposes methodologies to estimate them.
Previous works demonstrated that the *N*-REFB in the PBL could be recognized and estimated using
canonical transform approximations (Sokolovskiy, 2003) and could be reconstructed in the presence of ducting
conditions (Xie et al., 2006). Based on Xie et al. (2006), Wang et al. (2017) also showed an improved study based
on Xie et al. (2006) with an optimal estimation of negative bias using the provided precipitable water (PW) from
Advanced Microwave Scanning Radiometer for EOS (AMSR-E) microwave radiometer satellite data. Wang et al.
(2020) further proposed a bias estimation algorithm by generating a candidate set of modeled ducting profiles.
The one with the vertical gradient of the reflected bending angle closest to the observed profile is taken as the
bias-corrected profile. However, there are some limitations with these methods, such as that they only correct
ducting-related bias and information on the reflected bending angle is needed. For the RO observation error, the
local spectral width (LSW), which measures the uncertainty of the RO bending angle, has been used to indicate
the quality of the individual RO profiles. The LSW represents the errors caused by the nonspherical symmetry of
refractivity in the moist troposphere (Gorbunov, 2006). The LSW parameter has improved the use of RO
observations in data assimilation, including in the QC procedure (Liu et al., 2018) and dynamic estimation of RO
error in the lower troposphere (Zhang et al. 2022). Furthermore, Bowler (2020b) proposed estimating the RO error
with information on mean temperatures below 20 km, rather than using latitude to show meridional dependence.
In the presence of strong moist convection, nonspherical symmetry may cause rays to have the same impact
heights and increase the spectrum of the spectral components (Sokolovskiy et al., 2010). All these results suggest
that variations in LSW, temperature and humidity are directly related to the bias. Thus, we attempt to develop a
bias estimation algorithm that adaptively considers the uncertainty associated within each RO profile using LSW
and PBL thermodynamic variables such as temperature and water vapor.
In this study, we first investigate the characteristics of the FS7/C2 RO refractivity bias and establish
regression-based bias estimation algorithms. Two types of algorithms are examined. One is based on the physical
LSW parameter, and the other is related to the thermodynamic variables (temperature and water vapor). By
comparing the results of the estimated bias, we can identify the characteristics of each participating variable.
Finally, a bias correction method for the RO profile in the lower troposphere is proposed by combining the two
error estimation algorithms. We expect that this new algorithm can be used to improve the QC step and increase
the value of RO profiles in the lower troposphere.
The remaining paper is organized as follows. Section 2 provides the data information and methods for
estimating the refractivity bias. Section 3 discusses the general characteristics of bias and its sensitivities with
respect to different variables and land/sea conditions. Section 4 presents the results of bias estimation algorithms.
Finally, the summary and conclusion are provided in Section 5.



**2. Data and methodology**
**2.1 GNSS RO FS7/C2 and ECMWF data**
This study uses the FS7/C2 RO atmospheric profiles (atmPrf) processed by the Taiwan Data Processing Center
(TDPC) from Taiwan Data Process Center (TACC). The study period is from $1^{st}$ December 2019 to $29^{th}$ February
2020, before the FS7/C2 data were assimilated in the ECMWF analysis. All collected RO profiles are distributed
between 45°S and 45°N due to the inclination of the FS7/C2 satellite. A total of 244,853 profiles are collected
with the flag of "good data" during the periods, and only data below the height of 25 km are used to focus on the
bias characteristics in the troposphere. The data quality of the new FS7/C2 constellation is improved due to the
use of the advanced RO receiver and postprocessing with open-loop tracking. Most of the profiles show a
penetration improvement with depths below 1 km, and the penetration rate is 40% higher than those of FS3/C
(Chen et al., 2021c). Figure 1 shows the number of profiles that are retrieved when the radio ray penetrates below
the 1.5 km-height of sea level during the selected periods. The FS7/C2 data are mostly in tropical areas and have
more profiles penetrating below 1.5 km over oceans than over land.
For comparison, the reference RO profiles are calculated using the ECMWF atmospheric reanalysis (ERA5)
specific humidity and temperature. The RO refractivity bias (REFB) is defined as the mean difference between
the FS7/C2 and the ERA5 RO profiles (Eq. 1). In Eq. (1), $REF_i^{FS7/C2}$ is the $i^{th}$ RO refractivity profile, $REF_i^{EC}$ is
the reference profile, and $n$ is the total profile number.
$$\text{REFB} = \frac{1}{n}\sum_{i=1}^{i=n} REF_i^{FS7/C2} - REF_i^{EC} \tag{1}$$

**2.2 Negative refractivity biases (*N*-REFB) under super refraction**
This section provides an overview of the *N*-REFB that occurs in the PBL. Sokolovskiy (2003) discussed
the details of estimating these *N*-REFBs. Assuming the atmosphere is spherically symmetric under multipath
propagation and a typical moist troposphere, the impact parameter $a$ can be defined as
$$a = rn(r)sin\phi = const \tag{2}$$

where $n$ is the refractive index, $r$ is the radius from the center of curvature to the ray path, and $\phi$ is the angle
between the ray path and the radial vector. As shown in Tatarskiy (1968), the bending angle $\alpha$ of a GNSS-RO ray
path between two points $r^*$ and $r_0$ is given by
$$\alpha(r_0) = -2r_0 n(r_0) \int_{r0}^{r*} \frac{dn/dr}{n(r)\sqrt{r^2n^2(r) - r_0^2 n^2(r_0)}} dr \tag{3}$$

With (3), bending angle $\alpha(r)$ is the nonlinear function of refractivity index $n(r)$, and the convenient replacement
using $x = rn(r)$ and $a = r_0 n(r_0)$ transforms (3) into
$$\alpha(a) = -2a \int_a^{x*} \frac{d\ln n/dx}{\sqrt{x^2 - a^2}} dx \tag{4}$$

The bending angle $\alpha$ can be calculated as a function of impact parameter $a$ using (4). Under typical
atmospheric conditions, $dn/dr < 0$ and $dx/dr = n + rdn/dr > 0$. Under normal conditions, when refractivity
is spherically symmetric, the transformed RO signal is quasi monochromatic, with no bias introduced by additive



noise (Jensen et al., 2006). However, in the presence of a large vertical gradient, refractivity is nonspherically
symmetric, and noise appears because of multiple rays (Sokolovskiy 2010). However, if the refractivity is greatly
increased due to super refraction (SR), or $dn/dr < -n/r$ (or $\frac{dN}{dr} < 157\text{km}^{-1}$), then $dx/dr < 0$. The refractivity
within the SR layer is sufficient to trap the signal that carries the tangent point information (the geometry of GNSS
RO with the locations of the transmitter and receiver). In this case, when using Eq. (3) to calculate the bending
angle, assuming $r^* < r$, the term $rn(r)$ becomes less than $r^*n(r^*)$ due to the negative gradient of $x$ between the
top and bottom of the layer. This results in a negative sign contained within the square root of Eq. (3).
Consequently, the refractivity determined by the Abel inversion below the SR layer becomes negatively biased
(Sokolovskiy, 2003; Wang et al., 2020).
Under certain conditions, extreme SR occurs, and the signal is trapped within a strong and shallow
inversion layer. This is called the atmospheric duct. Ducting is more prevalent several kilometers above the stable
maritime atmosphere than over land. Previous works (Ao et al., 2008 and Feng et al., 2020) showed that areas
with cool sea surface temperatures, such as the eastern ocean, commonly have ducting. Atmospheric conditions
with a strong vertical lapse of humidity at the PBL top or temperature inversion are favorable for ducting, such as
evaporation ducts over warm SST and frontal inversion (Hsu, 1998). However, Wang et al. (2020) clarified that
evaporation duct cases would not introduce negative bias since the RO profiles are cut off at higher altitudes.
Notably, the *N*-REFB may not be completely attributed to the ducting effect. While *N*-REFBs on land are often
related to complex terrain, such as the high mountains of the Himalayas and North American Cordillera (Feng et
al., 2020), other *N*-REFBs over the oceans are located over the warm-moist Indian Oceans and Western Pacific.
This result means that parameters containing information under different conditions leading to REFB should be
examined.
This study employs different sets of variables to quantify the GNSS-RO REFB, including physical
parameters (LSW) and thermodynamic parameters (temperature and specific humidity). Each parameter attempts
to define different attributions of the observational error in GNSS RO data. Liu et al. (2018) used a linear function
of LSW/2 to illustrate the FS3/C dynamic error variance in the bending angle and refractivity. Following Liu et
al. (2018), we use the variable LSW/2 and modify this relationship to a polynomial regression. The other bias
estimation model is established using the thermodynamic variables to emphasize the impact of the thermodynamic
structure on REFB within the PBL.
**2.3 Algorithms for bias estimation**
Two types of regression models are developed to estimate the REFB. The first one uses LSW/2 as the
predictor, and the other uses temperature (T) and specific humidity (Q) as the predictors. Afterward, the regression
models are referred to as the LSW and TQ estimators, respectively. The LSW represents the RO inversion
uncertainty, and T and Q represent the impact of the thermodynamic structure on REFB within the PBL. Each of
these variables is expected to partly explain the characteristics of the bias. In each estimator, the order of the
polynomial and regression coefficients are optimized by using the R-square to assess the goodness of the fitting
ability. The data are subsets for training (80%) and testing (20%). To derive a robust fitting model, independent
fitting is performed five times by replacing the testing data with another 20% of the data. The regression model
with the highest score for both training and testing data is retained. According to the coverage of the FS7/C2 data,



we group the RO REF profiles from 45°S to 45°N into 5° x 3° boxes (Figure 1), and the estimators are built in
each box. In total, there are 72 x 30 boxes. The purpose of a region-dependent model is to improve the performance
of the estimator by considering the spatial variation in the REFB.

187         The optimal regression model for the LSW estimator is a second-order polynomial function. Eq. (5) shows

the formula of the LSW estimator in the $i_{th}$ box
$$u_i = \alpha_{i,1}x_i^2 + \alpha_{i,2}x_i + \alpha_{i,3} \qquad (5)$$

where $u_i$, the predictand, is the REFB, $x_i$ is the LSW/2, and $\alpha_{i,*}$ are the regression coefficients. It is expected that
the LSW reflects the issue of multipath propagation of the radio ray, and thus, this estimator quantifies the
relationship between the RO inversion uncertainty and REFB.
A similar procedure is applied to derive a multivariable polynomial regression model with T and Q as the
predictors. Here, both T and Q are obtained from the RO wet products after the one-dimensional variational
retrieval product. Before fitting, T and Q are standardized as
$$\chi = \frac{x_i - \min(x_i)}{\max(x_i) - \min(x_i)} \qquad (6)$$

where $\chi$ represents a normalized quantity ranging between 0 and 1 and $x_i$ is the original value of Q or T in the $i^{th}$
box. The optimal fitting model is
$$u_i = \beta_{i,1}y_i^2 + \beta_{i,2}y_i + \beta_{i,3}y_iz_i \qquad (7)$$

where $u_i$ is REFB, $y_i$ is the normalized Q, $z_i$ is the normalized T and $\beta_{i,*}$ are the regression coefficients.

201         We further apply the minimum variance estimation (MVE, Clarizia et al., 2014) to combine the results

from the LSW and TQ estimators. This approach has the advantage of having a smaller RMS error than either the
LSW or TQ estimation. The MVE is built to linearly combine the estimations so that the new estimation has the
minimum error variance:
$$u_{i,MVE} = \mathbf{m} \cdot \mathbf{u} \qquad (9)$$

where **u** is the vector of individual estimated refractivity bias and **m** is the vector of combination coefficients. One
of the advantages of this combination is that **m** is derived considering the error covariance matrix of individual
bias estimators.
$$\boldsymbol{m} = \left(\sum_{i=1}^{K}\sum_{j=1}^{K} c_{i,j}^{-1}\right)^{-1} \mathbf{C}^{-1}\mathbf{1} \qquad (10)$$

where **1** is a vector of ones, $K$ is the dimension of $\boldsymbol{m}$ ($K = 2$ in our application), $\mathbf{C}^{-1}$ is the inverse of the
covariance matrix between the individual estimation errors and $c_{i,j}^{-1}$ are the elements of $\mathbf{C}^{-1}$.
The element of the error covariance matrix $\mathbf{C}$ is expressed as $\boldsymbol{c_{ij}} = \langle (\boldsymbol{u_i} - \boldsymbol{u_t})(\boldsymbol{u_j} - \boldsymbol{u_t}) \rangle$, where $u_i$ is the $i^{th}$ bias
estimation and $u_t$ is the real bias.



## 3 Characteristics of the refractivity bias

### 3.1 General characteristics of REFB

Figure 2a shows the profile of the averaged REFB and its standard deviation from 0-25 km. RO data have significant $N$-REFBs in comparison to the ERA5-based RO reference, especially in the low troposphere. The bias is evident below 5 km and is largest at the surface with an amplitude of approximately -11 $N$-units. Given the large variations in moisture and temperature in the low troposphere, the standard deviation below the 2 km height increases as the height decreases. Notably, although the total number of profiles quickly decreases below 5 km (Fig. 2b), there remain enough data for near-surface statistical evaluation. The mean LSW (red line in Fig. 2a) also increases sharply as the height decreases, with two peaks at the surface and at the top of the PBL.

Figure 3a shows the latitudinal cross-section of the REFB. It is evident that the significant value of REFB below 5 km is primarily in the subtropics and tropics and slightly shifted to the Southern Hemisphere due to the austral summer. The opposite pattern, which has a high bias shifted to the Northern Hemisphere, is also seen with the data from June to August 2020 (not shown). This result indicates the general dependence of the distribution of $N$-REFB on the seasonal temperature structure. Similar to the $N$-REFB pattern, the large LSW is mainly exhibited over the tropics, tilting toward the Southern Hemisphere with the maximum near the surface (Fig. 3b). This finding illustrates that LSW variation can be related to the REFB to some extent. Moreover, other high LSW values are located a few kilometers above the surface of the Southern Hemisphere. Under summer conditions, the large lapse of humidity on the top of the moist PBL leads to a strong vertical gradient of refractivity. A similar pattern is also found in the study of Zhang et al. (2023) but with the FS3/C data in August 2008. The increased LSW just above the boundary layers could be caused by common inversion layers in the troposphere of some oceans. Another effect that could be considered is the influence of convective clouds just above moist oceans (Yang et al., 2016). The large LSW near the surface in Fig. 3b reflects the ability of FS7/C2 to penetrate deep into the moist troposphere of the tropics, which was not seen in Zhang et al. (2023).

### 3.2 Dependence on geography and thermodynamic conditions

As the REFB has seasonal dependence, we further examine the dependence of the REFB on land/ocean and thermodynamic conditions. Figure 4 shows the general comparison of REFB between land and ocean, together with its standard deviation (stdv) and LSW. Over ocean, both REFB and LSW below 4 km are larger than those over land, and the $N$-REFB extends to higher altitudes (Fig. 4c vs. 4d) with a greater vertical gradient of REFB below 2 km. The magnitudes of mean REFB and stdv above 2 km are comparable over land and over ocean. The LSW over ocean below 4 km increases faster over the ocean, and the second peak value at the PBL top is much larger. Therefore, the REFB varies differently over land and oceans, and the LSW exhibits similar sensitivity. This feature suggests the potential of LSW as a predictor for estimating $N$-REFB to account for the difference between land and ocean. Notably, the number of RO profiles over land is about 21% of the total profiles, and the penetration rate is lower than the RO profiles for ocean (Fig. 1). This finding may contribute to a larger stdv over land below 1 km.

Given the large REFB near the surface, we focus on the regional variations in REFB within the PBL. Figure 5a clearly shows that the $N$-REFB below 1.5 km is large over the ocean, particularly over the ocean off the western





coasts of the American and African continents. Small *N*-REFBs appear over the tropical Pacific and land. However,
there are small but positive REFBs over the high mountain regions. The different behavior of the *N*-REFB over
ocean and land implies the impact of regional variability and the associated thermodynamic structure in the PBL.
Furthermore, a large LSW usually corresponds to the region where the vertical gradient of refractivity is large,
which is attributed to the nonspherically symmetric irregularities of the atmosphere. This effect is expected to be
strongly associated with the large variation in the thermodynamic structure. We note that the temperature pattern
over the ocean in Fig. 5d is similar to SST and thus can represent the sea surface condition. As shown in Fig. 5b-
5d, high LSW occurrence is mainly located over warm-moist oceans, such as the equatorial Pacific Oceans,
equatorial Atlantic and Indian Oceans. However, not all of the regions with high temperature and moisture coexist
with the regions with high LSW. Some exceptional regions can be seen, such as offshore to the coast of Southwest
Australia and offshore of Southwest Africa.  Fig. 5 suggests that although LSW, temperature and specific humidity
have certain cross-relationships, the characteristics of thermodynamic conditions cannot fully explain the
distribution of LSW. In other words, a REFB estimation model, which is based on only one variable, is not enough
to explain REFB since their variation is different for some specific regions.
To further highlight the characteristics of REFB under different conditions, the REFB profiles are grouped
according to each profile's LSW, temperature and specific humidity averaged below 1.5 km for land and ocean
(Figure 6). In general, it is evident that the larger *N*-REFB increases with increasing LSW below 4 km, as shown
in Fig. 3; however, the characteristics are different for land and ocean. Over land, the very high LSW does not
guarantee the occurrence of a large *N*-REFB near the surface. Instead, *N*-REFB appears at the PBL top, and the
REFB turns positive near an altitude of 8 km. These REF profiles are near the coasts of North America and North
Africa. Moisture and temperature likewise exhibit the same linear relationship with *N*-REFB in the lower
troposphere. However, *N*-REFBs also tend to occur under conditions of low moisture over the ocean. Figure 6
reveals that the relationship between REFB and LSW, T and Q under 1.5 km is dominantly linear; however, the
REFB variations can be further explained by a quadratic relationship with T and Q.

## 4 Results of bias estimation

### 4.1 General performance

In this section, we present the estimation for REFB using the methods introduced in Section 2. As mentioned,
LSW/2, which represents the retrieval uncertainties of the bending angle and, hence, refractivity uncertainties, is
the predictor for the first bias estimation model. The temperature and specific humidity retrieved from FS7/C2
RO data are the predictors for the second estimator. Although the T and Q products retrieved from RO profiles
are not as optimal as those retrieved from other analysis products, they still provide valuable information to
estimate the real bias through the training process, as described in Section 2. In the following section, we examine
the general behavior of the estimated *N*-REFB as a function of each predictor set: LSW and TQ.
Figure 7 shows the relationship between the REFB and LSW/2 for the Southern Hemisphere (SH) during
the study period. Here, we focus on the austral summer in the SH, which could emphasize the warm and moist
conditions in our study period. In Fig. 7, REFB is grouped every 2% of LSW/2, from 0 to 36%. The solid and
dashed lines show the LSW-based *N*-REFB estimates for ocean and land, respectively. Under 1.5 km, the



magnitude of the *N*-REFB as a function of LSW is much larger for oceans than for land. Generally, as LSW/2
increases, the REFB becomes more negative below 1.5 km for both land and ocean. The correlation for data below
1.5 km is 0.94 for oceans and 0.9 for land with the training data. As shown in Table 1, the correlations over ocean
and land are robust and similar to the training and testing data. We note that the positively proportional trend is
not evident for the data above 1.5 km, and there is little difference in *N*-biases between land and ocean.
Figure 8 shows the result of the second bias estimator, which relates the REFB with temperature and
specific humidity (TQ) for the SH under 1.5 km. The TQ estimation over ocean and land can capture the feature
where the REFB becomes more negative under moist conditions. Similar to the LSW estimator, the TQ estimator
shows a stronger dependence over the ocean. As shown in Fig. 8a, given a fixed specific humidity, the relationship
between REFB and temperature is parabolic under moist conditions but linear under dry conditions. As the water
vapor increases, the estimated REFB tilts toward lower temperatures (e.g., the minimum of estimated REFB
appears at 22.8°C when Q is fixed at 5g/Kg, but it appears at 15°C when Q is fixed at 15 g/Kg). This finding
reflects the condition over the cool SST, west of the coast of South America and South Africa. Over land, there
are fewer data with large negative REFBs. In addition, the estimated REFB gradually tilts toward positive values
as the water vapor decreases, which is associated with the dry conditions over the mid-latitude continent (Fig. 5c).
The multivariable regression has a high correlation coefficient equal to 0.79 and 0.72 for ocean and land,
respectively. Thus, the result also suggests that T and Q are suitable for use to estimate the refractivity bias. Figure
7 and Figure 8 confirm that models with LSW/2 or TQ as predictors can estimate the REFB under 1.5 km, but
there are different sensitivities for ocean and land.
In the next step, we further apply these regression methods to construct the region-dependent bias
estimation model using the data in a 5° × 3° box within 45°N to 45°S. The estimators are built for each box to
represent the regional variation pattern of *N*-REFBs.
Figure 9 shows the horizontal distribution of the mean real and estimated REFBs with the training and
testing data. Notably, there are some differences between the training and testing data (Fig. 9a vs. 9b), such as the
large REFB off the coast of Australia. In comparison to the real REFB distribution (Fig. 9a), the LSW-based
REFB (Fig. 9c) captures the general pattern with larger biases over ocean and lower biases over land in both the
training and testing data. However, the LSW-based REFB is less capable of capturing the large bias over the
subtropical oceans off the west coast of South America and South Africa and Australia. Those are expected to be
the oceans that have a cold SST, where ducting and SR occur commonly due to the frequent occurrence of
inversion layers on top of the surface cold atmosphere. Although the LSW-based REFB can also represent a
portion of the *N*-REFB in these regions in general, it is obvious that the values are underestimated there. The
LSW-based estimation exhibits good performance in estimating the *N*-REFB in the Indian Ocean, where the
pattern and magnitude of the estimated REFB are close to those of the real REFB. In contrast to the LSW-based
REFB, the TQ-based REFB represents the large *N*-REFB in the high-ducting-occurrence regions well. Although
the magnitude of the *N*-REFB offshore the coasts of South America and South Africa is still underestimated, the
pattern and amplitude of the *N*-REFB are much better represented in comparison with the LSW-based estimation.
In addition, the TQ-based estimation captures the low bias pattern well, such as the tropical western Pacific,
western South America and Africa, while the LSW-based estimation overestimates the negative bias. The similar
pattern between the real and TQ-based estimated *N*-REFBs can be explained by two reasons. The first reason is




the ability to capture SST characteristics. For example, cold SST regions can result in a cool, low moisture near-
surface atmosphere (Fig. 5c and 5d) and impact the boundary layer. Second, the bias in the RO profiles will be
translated to the retrieval products, which makes the predictors highly related to N-REFB in the ducting areas.
This finding also confirms that the N-REFBs below 1.5 km are highly related to the thermodynamic conditions
and that the TQ estimation successfully reflects the impact of the air-sea interaction on the RO refractivity.
The third method, the MVE, combines the two independent estimations. As described in Section 2, the
MVE derives the optimal combination by considering the error correlation between the individual estimations.
This method has the benefit of having an RMS error that is less than the lowest RMS error in each bias estimate
and thus could inherit the benefits of each estimation. Notably, the MVE approach requires knowledge of the error
covariance matrix between two components (Eq. 9). The error correlation of the two REFB estimators is 0.294.
Normally, the high error correlation indicates the dependency between two components and thus less benefit from
using the MVE method. Although LSW is known to have a relationship with tropospheric water vapor variation,
our experimental results indicate that the error correlation between two estimates is low enough that it is expected
that the MVE can extract useful information from both estimations. Compared to the LSW and TQ bias estimation,
the results of the MVE showed a pattern closer to the real REFB with both the training and testing data sets. This
finding confirms that the MVE N-REFB carries the advantage from individual estimators. For example, the MVE
REFBs can show the high N-REFB in subtropical oceans off the west coast of South America, South Africa and
Australia from the TQ-based estimation, and it can avoid underestimations with the LSW estimator.
Simultaneously, the MVE REFBs avoid the overestimation of N-REFB offshore of the western coast of North
America and the southern Pacific shown in the TQ-based REFBs due to information from the LSW estimation.
To confirm the performance of the bias estimation, we further compute the root-mean-square error
(RMSE) between the real and estimated REFB in each box. Figure 10 clearly shows the contribution of each
estimation in estimating bias for land and oceans and reflects the representativeness of the mean REFB shown in
Fig. 9. Almost all the large RMSEs in the LSW or TQ estimation are removed by the MVE method (Fig. 10c and
10f). The LSW-based estimation exhibits high RMSE in the cold SST regions and several ocean regions, such as
the Southeastern Atlantic, Southeastern and North Western Pacific Oceans, while the TQ estimation successfully
mitigates this issue. On the other hand, the LSW-based estimation performs better in the tropical Atlantic and
Indian ocean. Again, the MVE estimation has the smallest RMSE compared to the other two estimations,
especially over oceans. With the testing data (Fig. 10d-10f), the RMSEs become larger in individual estimations,
as expected. Most importantly, the MVE method retains its advantage in the optimal estimation, with an RMSE
smaller than that of either estimation. In other words, the REFB can be better estimated by considering the
characteristics in different predictors. Table 2 shows the global mean RMSE. The TQ method has a smaller RMSE
compared to the LSW estimation. The MVE method further improves the TQ method by 32% and 23.6% with the
training and testing data, respectively.
**4.2 Verification and Calibration**
This section examines the performance of the REFB estimation methods and whether they can be used for
calibrating the refractivity profiles. Taking two areas (indicated in Fig. 9a) with different REFB characteristics
as examples, the REFB profiles are grouped by an interval of 0.5 km in the vertical direction. Area A is in the



region of Eq < Lat < 10°N and 55°E < Lon < 75°E, and Area B is in the region of 20°S < Lat < 30°S and 105°W < Lon < 85°W. For each area, the estimated REFB at different levels are derived using the same estimation methods defined in the previous section. Figure 11 shows the mean of the real and estimated REFB profiles in two areas with the testing data. We note that the results of the training and testing data are very similar. The general pattern of the REFB profiles reflects the characteristics of the atmospheric conditions in that region. In Area A, the mean N-REFB is large at the surface but gradually decreases to zero at the 3-km height. In this case, the air below 2 km is very warm and moist over the Indian Ocean (Fig. 12). The highly humid condition gives a large LSW (Fig. 6b), and thus, the LSW method can have a good ability to estimate bias in this circumstance, while the TQ method overestimates the $N$-REFB. In contrast, Area B shows different patterns (Fig. 11b): the real $N$-REFB is even larger (-17 N) at the surface, and the negative bias at 2 km is still large compared to that in Area A. As shown in Fig. 12, this characteristic is associated with the inversion layer at 2 km over the cold SST region and large vertical moisture gradient, a typical condition of ducting. While the LSW-based estimation underestimates the N-REFB with the existence of the inversion layers this can be captured by TQ-based estimation. Nevertheless, the MVE method is always much closer to the real REFB, as it utilizes the advantages of each of the individual estimates.

We further examine whether our MVE estimations can capture the behavior of the REFB profiles in these areas. To effectively illustrate numerous real and estimated REFB profiles, we group them into different bins of bias and present the results in terms of probability. In Figure 13, each bin spans 0.6 km height and 3 $N$. The comparison of the probability distribution is performed with the training data due to the limitation of the samples. In general, the real REFB probability in Area A has a broad distribution. The distribution is skewed to a large negative bias near the surface but skewed slightly to a positive bias above the PBL at altitudes of 3 to 5 km. The estimated REFB profiles exhibit similar behavior, including the positive bias above the PBL. Compared to Area A, the real REFB probability of Area B is more skewed near the surface. The spread quickly decreases as the altitude increases and skews slightly toward a positive bias at the 2-km altitude. Such a characteristic is attributed to the fact that Area B is in the ducting region where the cool stable PBL confined the fluctuation of bias. The behavior is also well captured by the estimated REFB profiles. The results in Fig. 13 suggest that the mean bias is well represented by the bias estimation method, and the statistical distribution of the estimated REFB is also consistent with the real REFB. As expected, bias estimation can be applied to calibrate the RO refractivity profiles.

**4 Conclusions**

This study investigates the characteristics of refractivity bias (REFB) of FS7/C2 and its sensitivities to RO measurement uncertainty (LSW) and thermodynamic conditions (temperature and moisture). With the optimal purpose of calibrating REFB, two bias estimation models are constructed based on polynomial regression with the LSW, and temperature and specific humidity are used as predictors in each estimation. The study period is the winter of 2020, with the ECMWF reanalysis data taken as the reference truth.

Similar to previous studies, the low-level FS7/C2 RO refractivity data of during the study period still contain significant bias when compared with ECMWF reanalysis data. In general, the REFB below 1.5 km is negatively proportional to LSW and exhibits a stronger dependency over ocean than over land. However, it is noted that high



LSW over land does not guarantee the occurrence of a large REFB. Additionally, REFB in the PBL has a strong
dependence on low-level temperature and moisture. While the majority of Pacific and Indian Oceans with warm
SSTs have significant $N$-REFBs, the largest $N$-REFB regions are near the cold SST regions off the western coasts
of South America and South Africa. Small and even positive REFBs are observed over South America and South
Africa.
Two REFB estimation models based on the polynomial regression approach are first applied to construct the
region-dependent REFB in the PBL (below 1.5 km). One estimation model uses LSW, and the other uses
temperature and moisture (TQ) as predictors. The estimation models are applied to 72×30 boxes from 45°S to
45°N. Furthermore, the MVE method is used to combine two REFB estimations. The results show that the bias
estimation models with either LSW or TQ have their own advantages in estimating REFB. The LSW-based model
shows the ability to capture the general pattern of N-REFB but significantly underestimates the N-REFB in the
ducting areas. On the other hand, the TQ-based model has great performance in representing the pattern and
amplitude of REFB, particularly the large $N$-REFB in the ducting areas and small REFB over most land regions.
The MVE estimation successfully adopts the advantage from either LSW or TQ estimation. Among the three
REFB estimations, the MVE model has the smallest RMSE. Three REFB estimation models are further applied
to reconstruct the REFB profiles. Both the LSW and TQ estimations can well represent the vertical gradient of
the mean REFB and the MVE estimation gives an estimated REFB profile closest to the real REFB with the
probability distribution similar to the distribution of real REFB. Therefore, our results suggest that the MVE
method can be used to calibrate RO refractivity profiles.
We should note that the methodology proposed in this study still has limitations. For example, the
temperature and moisture from the ERA5 reanalysis may have their own biases, and thus, the simulated
refractivity profiles could carry the bias as well. Therefore, we can only claim that our bias estimations are close
to the bias in which ERA5 is taken as the truth. In addition, factors such as temporal variations, local topology
and meteorological effects, are neglected in this study. The systematic bias may have more characteristics
regarding smaller scales spatiotemporally. For future work, bias estimation models will be constructed at higher
resolutions with more RO profiles collected from the current FS7/C2 or other operational and commercial GNSS-
RO satellites. The corrected refractivity can further add value to RO data in the PBL studies, such as improving
the low-level moisture analysis through data assimilation or improving the accuracy of the RO retrieval products
of temperature and moisture to expand their applications in PBL studies.
**Author contribution**: SY was in charge of the conceptualization of this study. SY and GP prepared the manuscript
with contributions from all co-authors. GP constructed the packages of bias estimation. SY and GP analyzed the
data. SY and GP wrote the manuscript draft; CC, SC, and CH reviewed and edited the manuscript.
**Competing interests**
The authors declare that they have no conflict of interest.
**Acknowledgments**
This work is supported by the Taiwan National Science and Technology Council grants NSTC-111-2121-M-008-
001 and NSTC-111-2111-M-008-030 and Taiwan Space Agency grant TASA-S-110316.



**Code and data availability**

The codes of the bias estimators used in this study are available at Github (https://github.com/jiajia170801/bias_estimation_paper). The RO data is obtained from TDPC (TACC) by https://tacc.cwb.gov.tw/data-service/fs7rt_tdpc/. The ECMWF reanalysis v5 (ERA5) data is obtained from Copernicus server by https://cds.climate.copernicus.eu/cdsapp#!/dataset/reanalysis-era5-pressure-levels?tab=overview.

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



**Table 1: Correlation coefficients between the real and estimated REFBs over ocean and land**

| Correlation coefficients | LSW based | | *T*Q based | |
|:---:|:---:|:---:|:---:|:---:|
| | ocean | land | ocean | land |
| **Training data set** | 0.94 | 0.9 | 0.79 | 0.72 |
| **Testing data set** | 0.93 | 0.89 | 0.71 | 0.70 |


**Table 2: Global mean RMSE of each REFB estimation in comparison to the real REFB**

| Global mean RMSE | LSW-based | *T*Q-based | MVE |
|:---|:---:|:---:|:---:|
| Training data set | 2.033 | 1.614 | 1.088 |
| Testing data set | 2.815 | 2.266 | 1.731 |





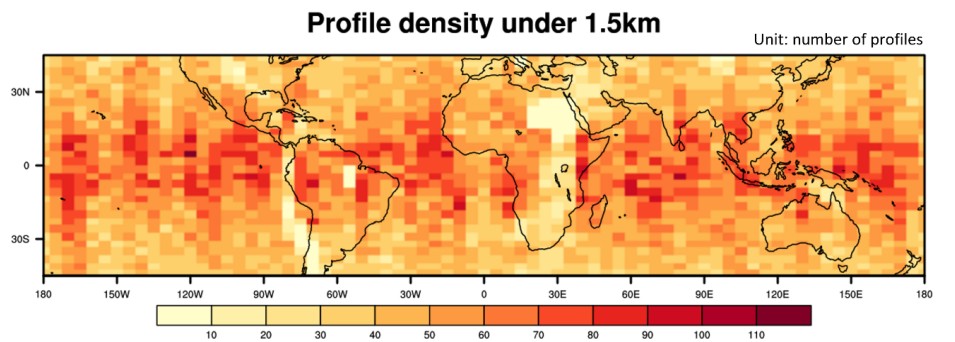


**Figure 1: Density of FS7/C2 RO profiles below the 1.5 km height during the study period (unit: number of profiles).**






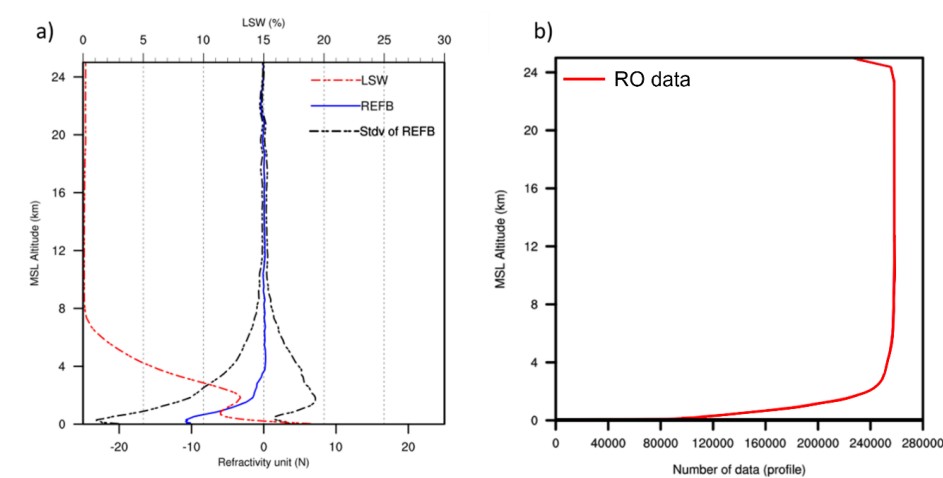

**Figure 2: (a) Mean and standard deviation of REFB and mean LSW during the study period. (b) The amount of available RO data during the study period (red: bending angle, blue: refractivity).**




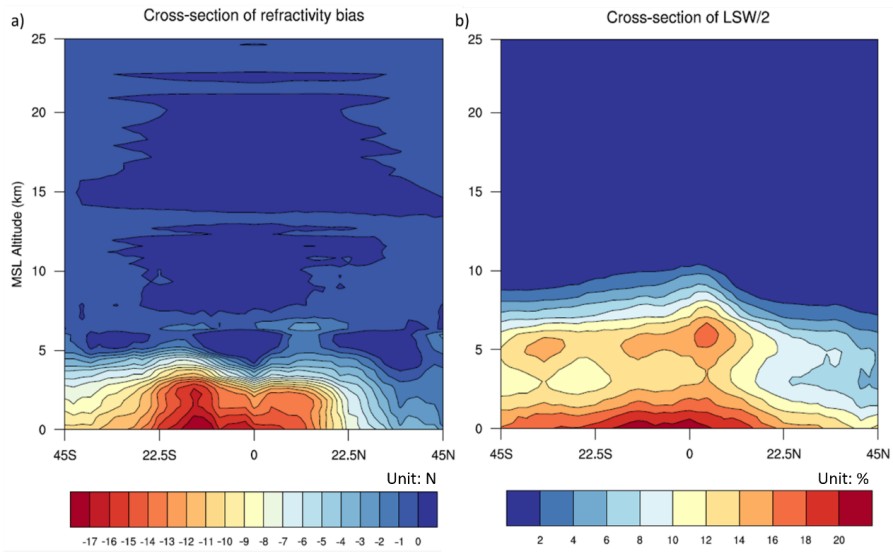

**Figure 3: The cross-sections of (a) mean REFB and (b) mean LSW/2 during the study period.**





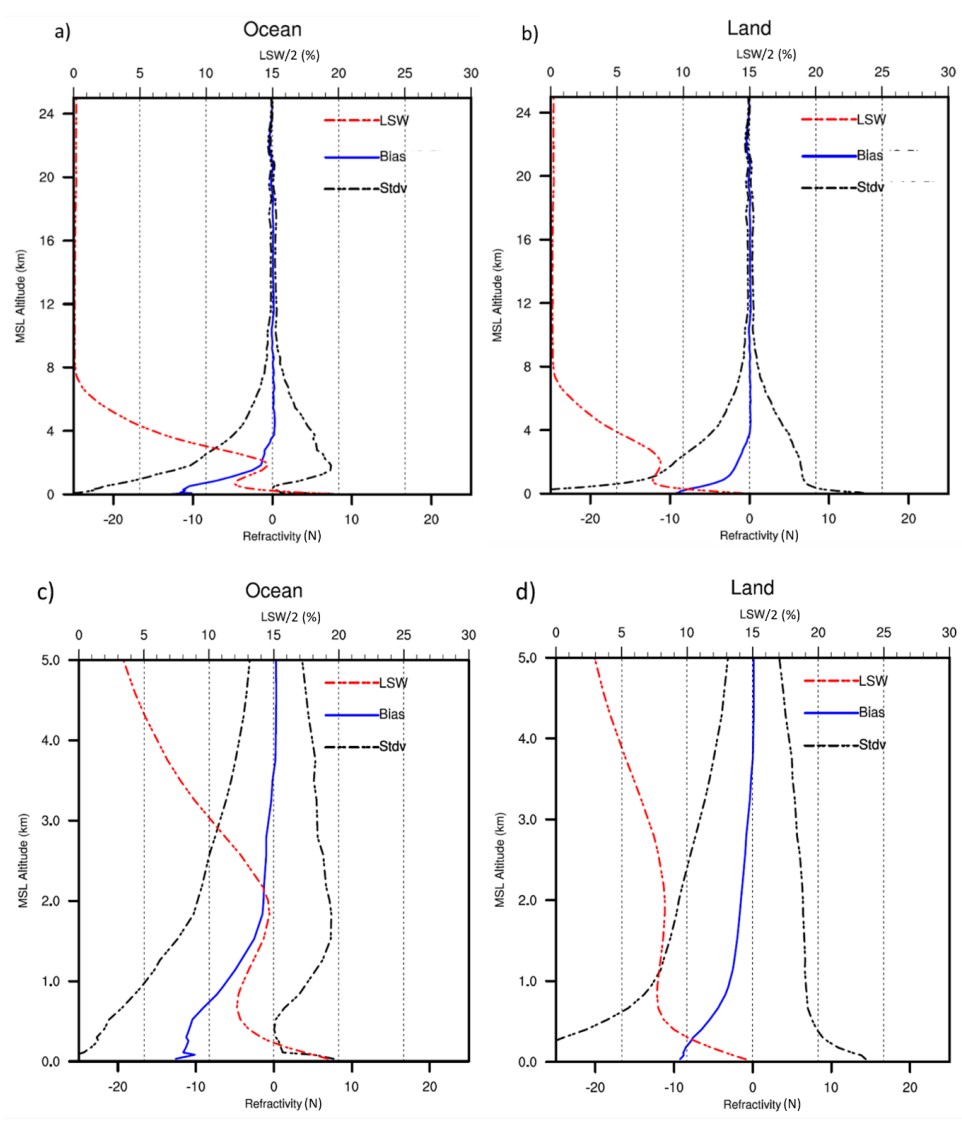


**Figure 4: (a) and (b) are vertical profiles of the mean ($N$) and standard deviation ($N^2$) of REFB, and mean LSW with**

**altitudes up to 25 km over ocean and land, respectively. (c) and (d) are the same as (a) and (b) except zoomed versions**

**below 5 km.**


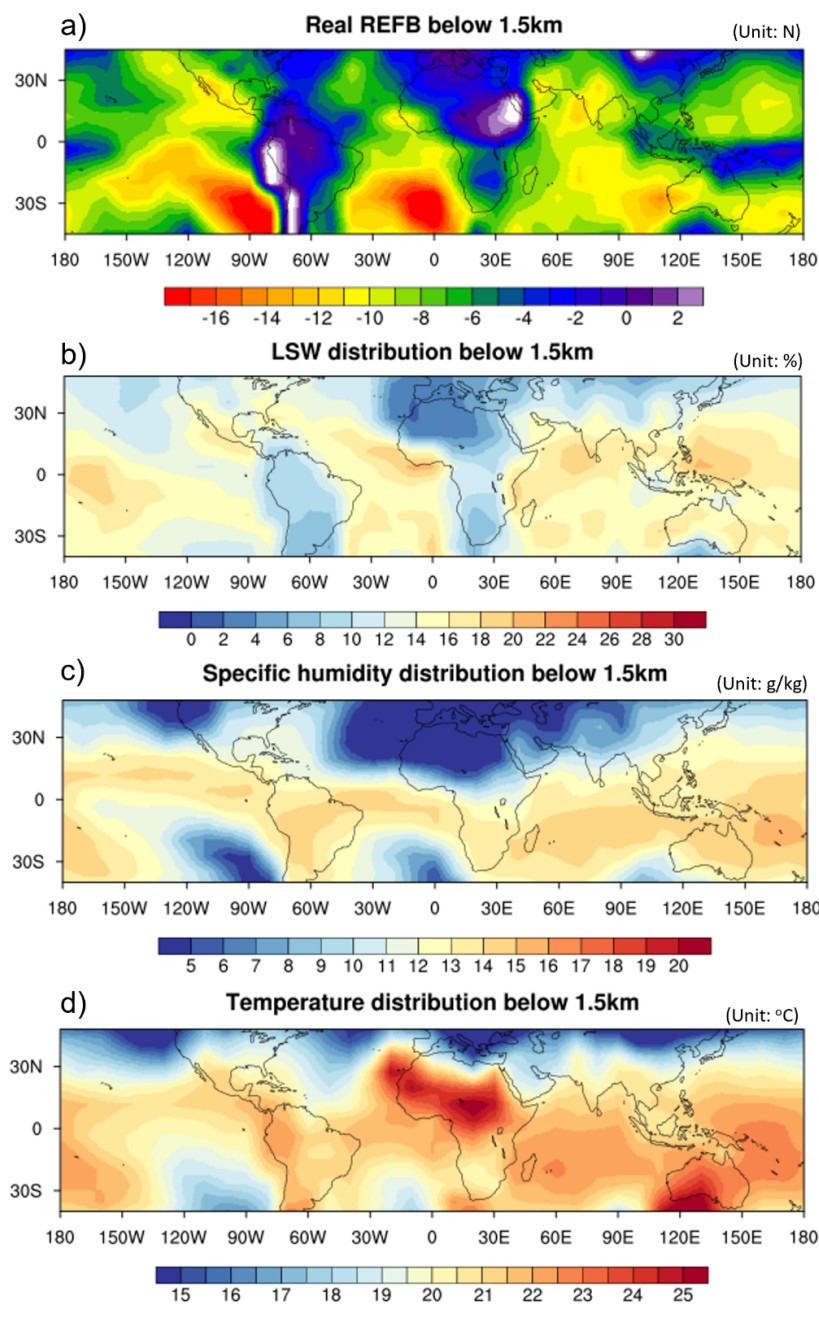

**Figure 5: Horizontal distribution of (a) REFB (*N*), (b) LSW (%), (c) specific humidity (g kg$^{-1}$), and temperature (°C) averaged during the study period.**



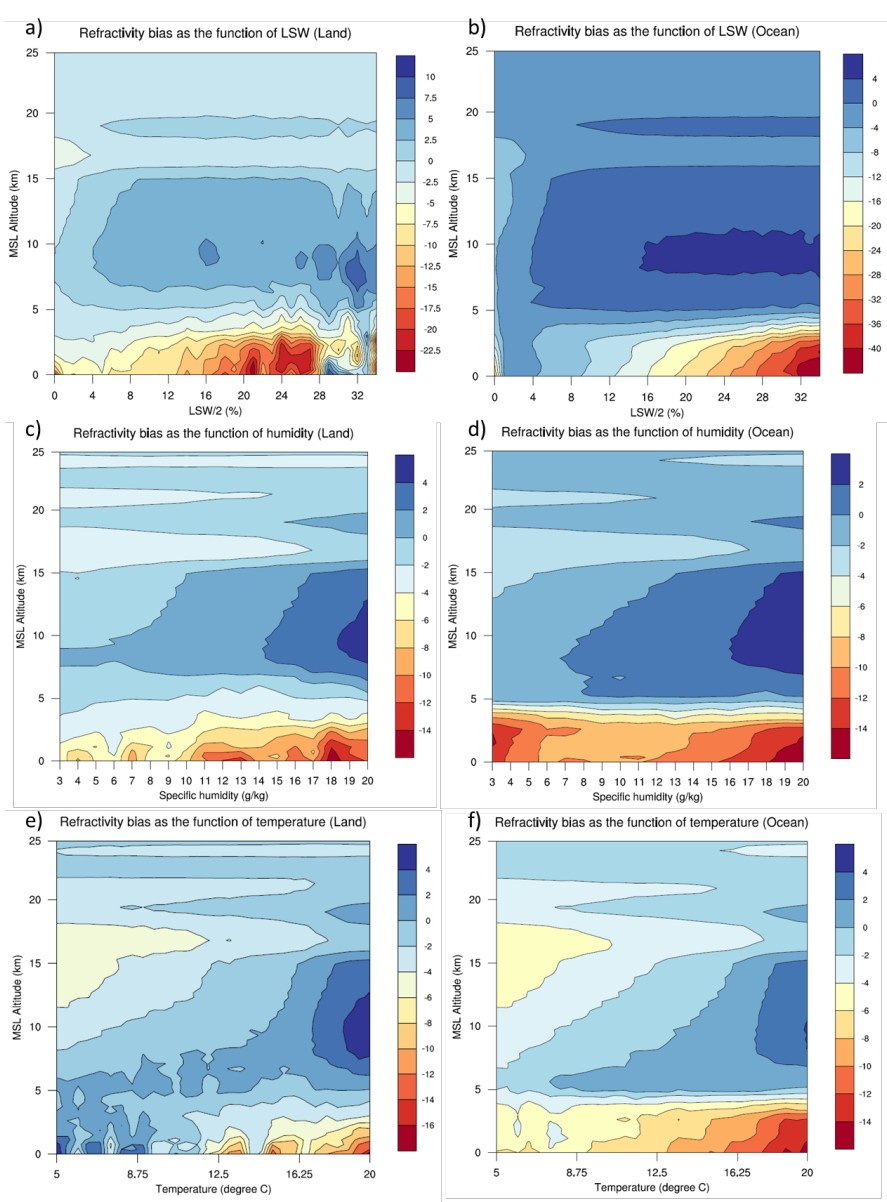

599

**Figure 6: Vertical cross-section of refractivity bias over the ocean as a function of height and (a) LSW/2, (c) specific humidity and (e) temperature over land. (b), (d) and (f) are the same as (a), (c) and (e), except over the ocean.**

602





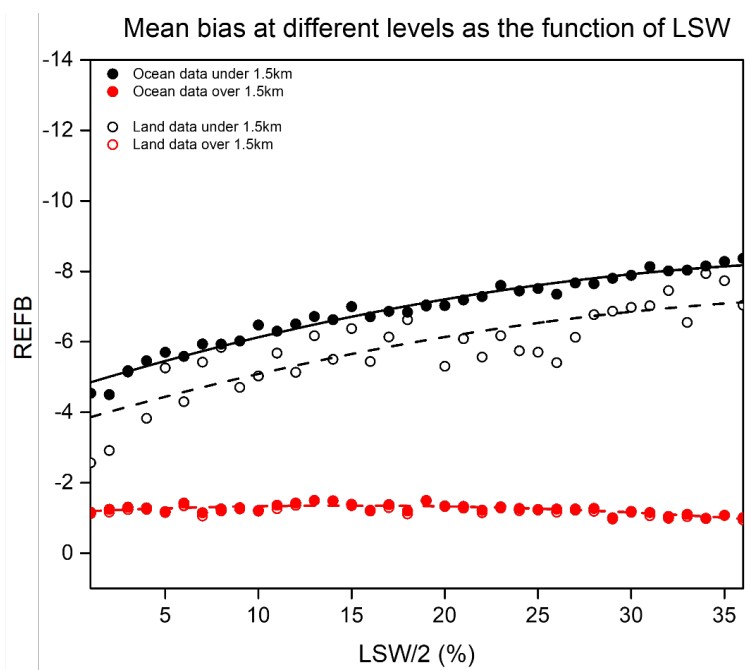

**Figure 7: Relationship between LSW/2 and REFB. The solid and dashed lines represent the N-biases computed model for the ocean and land, respectively, as a function of LSW/2 (Southern Hemisphere only).**



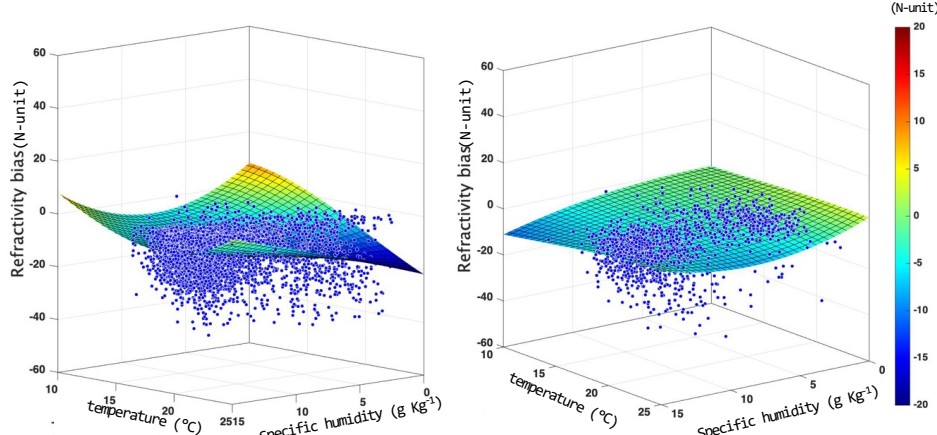

608

609 **Figure 8: Relationship among temperature, specific humidity and REFB for the Southern Hemisphere.**

610



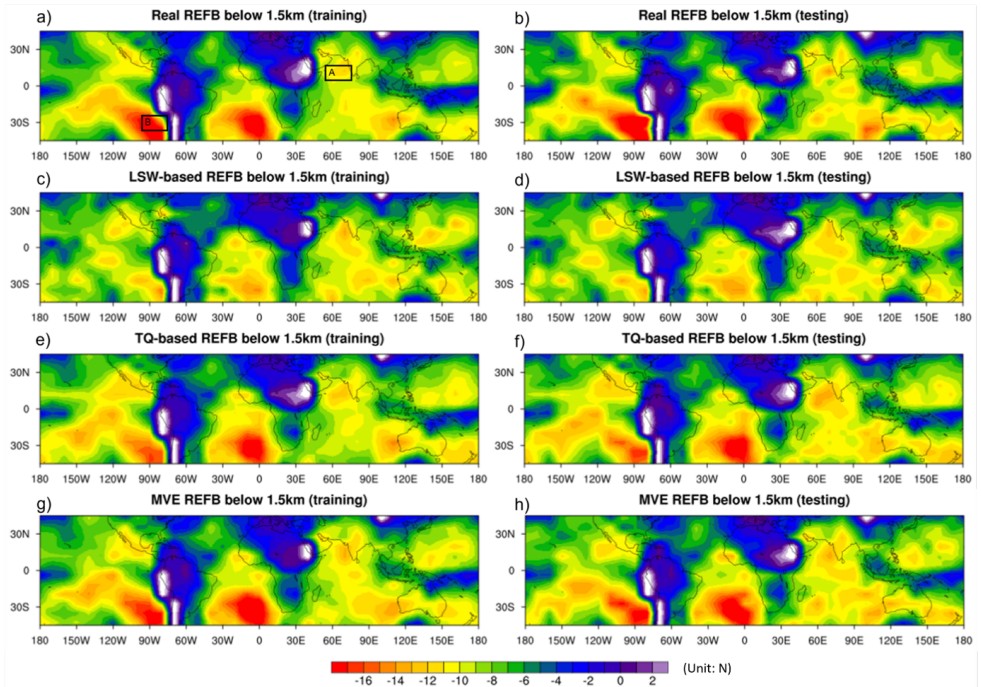

**Figure 9: Horizontal distribution of refractivity bias and different estimated refractivity biases. The boxes denoted A and B are the example boxes used in Figures 12 and 13, respectively.**




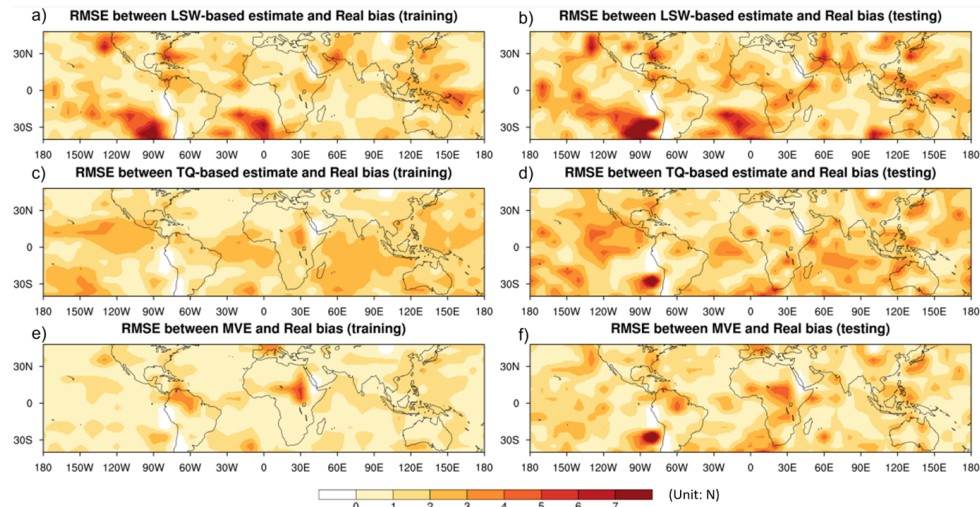


**Figure 10: Horizontal distribution of RMSE between the real REFB and estimated REFB by different methods with training (left column) and testing (right column) data.**





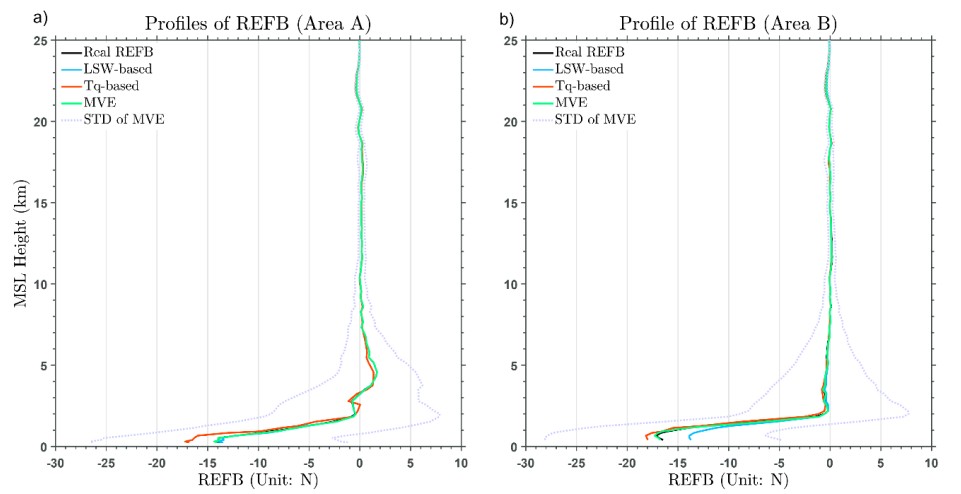

**Figure 11: Profiles of refractivity bias (real and estimates) for two different areas selected in Fig. 8a. Boxes A and B are in (Eq < Lat < 10°N, 55°E < Lon <75°E) and (20°S < Lat < 30°S, 105°W < Lon < 85°W).**



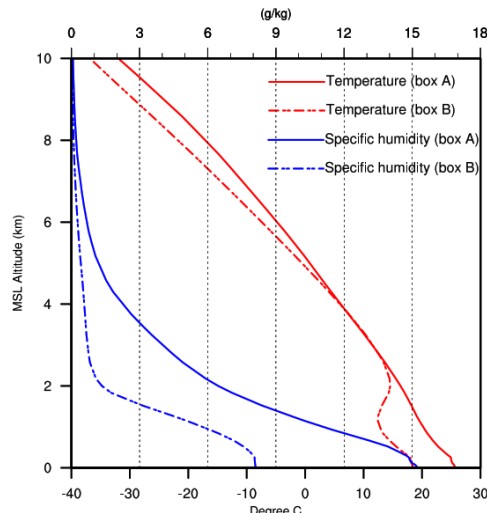


**Figure 12: Vertical profiles of averaged temperature (red lines) and specific humidity (blue lines) for Areas A (solid lines) and B (dashed lines).**






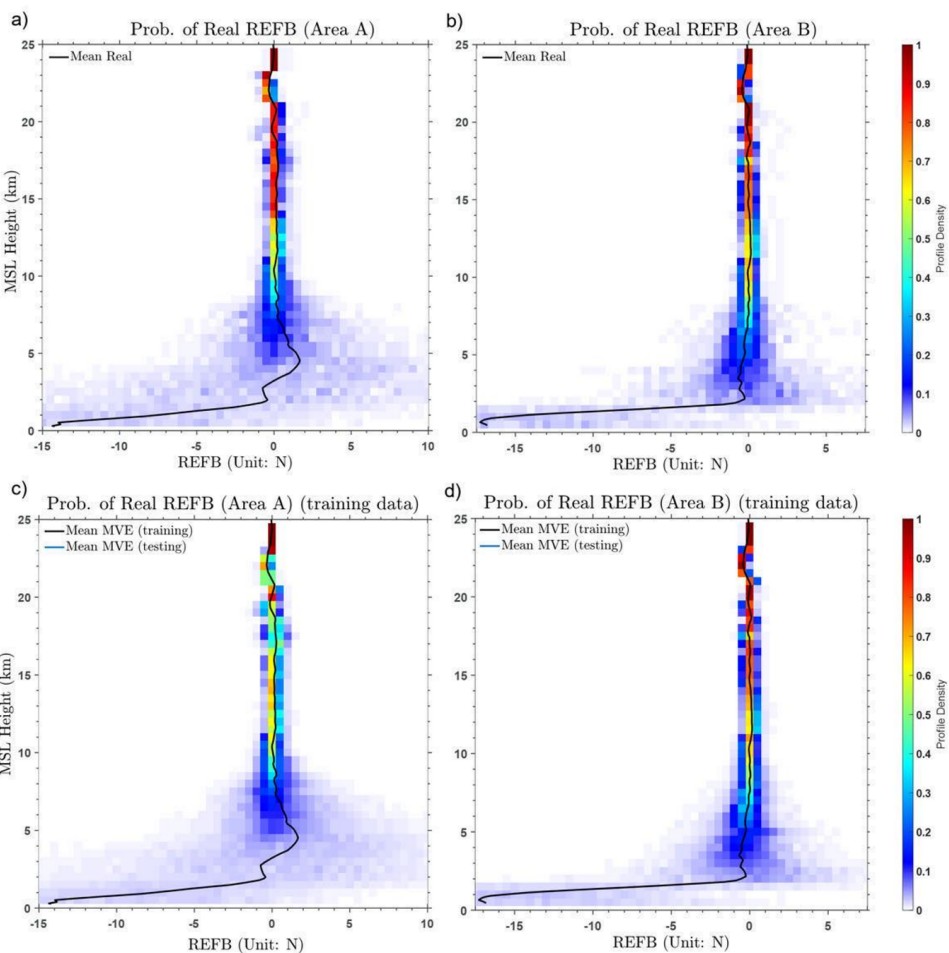


**Figure 13: Profiles of (a) real and (c) MVE REFB probability for Area A. The black line shows the mean MVE REFB**
**profile. (d) and (d) are the same as (a) and (c) except for Area B.**
