# Peer review of "Estimating the refractivity bias of Formosat-7/COSMIC-II 3 GNSS Radio Occultation in the deep troposphere"

_EGUsphere, 2023_

## Referee Comment (RC2)

Review for Pham et al., "*Estimating the refractivity bias of Formosat-7/COSMIC-II GNSS Radio Occultation in the planetary boundary layer*"

Summary

The refractivity retrieval bias in the planetary boundary layer has long been recognized and is an important issue that impede the GNSS radio occultation data assimilation in the lower troposphere. This paper developed the polynomial regression model to estimate the region-dependent biases with the assumption that the refractivity biases (refer to REFB in the paper) are related to the local spectral width (LSW) and the thermodynamic information including the temperature and specific humidity (T &q). An individual regression model with LSW and T & Q were derived. In addition, the minimum variance estimation (MVE) was applied to combine all three parameters into the regression model. The authors demonstrated that the MVE regression model reproduce the refractivity biases better than the individual models. The authors also claimed that the model can help correct the refractivity bias in PBL and thus improve the quality control of the GNSS RO and increase the values of RO observations in the lower troposphere.

Overall, this paper is generally well written, and the regression model estimation effort offer some improved understanding of the refractivity biases in the lower troposphere. However, some statement regarding the refractivity biases in the PBL were not accurate. The relatively poor performance of the LSW regression model over ducting region is not a totally surprise, as LSW might not ducting-induced biases is primarily caused by the Abel inversion (i.e., a singularity problem in retrieval algorithm). However, the physical explanation of the better relation between REFB over ducting region with the T&Q was not clear and need better explanation.

Moreover, it needs to be cautious that the 1D-Var retrieved temperature and humidity retrieval embed the uncertainty from model a-priori information. Therefore, it is susceptive that the use of such data in the MVE regression model could introduce solid positive impact to the data assimilation.

I would recommend "major revision" for the paper. The major comments along with some detailed technical comments are listed below:

Major comments:

1. The planetary boundary layer has been raised in numerous place in the paper, including the paper title. However, there are no clear definition of PBL, instead of a fixed height of 1.5 km as the indication for PBL. Note that the PBL height has various definition and can change significantly from a few hundred meters up to 3-4 km. It might be too much simplification for use one fixed height of 1.5 km to indicate the PBL.
2. The authors need to clearly stating that the Local spectral width, Temperature (T) and specific humidity (Q) used in this paper are the "average" below 1.5 km?

a. L267: Temperate and specific humidity averaged below 1.5 km for land and ocean (Figure 6)
  i. Do the authors considering the regional difference in penetration depth of RO profiles (Fig. 1)?
  ii. How to deal with the potential negative bias in T/Q resulting from the early termination of the RO profiles, i.e., missing the lowest part of profile below 1.5 km where temperature and moisture values are likely larger closer to the surface.
  iii. Do the authors process all RO data or only the ones penetrating certain threshold (1.5 km??)? Sensitivity test might be needed, for example, regenerate Fig. 5 by requiring RO profiles to penetrate below 500m above mean-sea-level. Or simply calculate the average between 500m and 1.5 km.

3. I am not totally convinced that the temperature and specific humidity patten correlated to the refractivity bias.
  a. The seemingly high correlcation between temperature/specific-humidity (T/Q) and the refractivity biases over the ducting regions is not a surprise, as those regions are mostly over the ocean upwelling region with cool sea surface temperature and relatively low moisture in PBL (below 1-2 km). However, the T/Q relationship to the REFB might not hold over high latitude.

L67-70:
- The authors give the impression that the "strong refractivity gradient" are the primary problem or RO measurement uncertainty. The authors claimed that the strong vertical refractivity gradient will introduce multipath and resulting complex bending angle and complexity of RO uncertainty. Also the L70: "strong refractivity" (I assume it should be strong refractivity gradient) causes a negative refractivity bias (N-REFB) (Rocken et al., 1997)
- However, the negative bias in lower troppposphere in earlier generation RO missions (e.g., Rocken et al., 1997) were mainly due to the close-loop tracking issue in early generation of RO receivers, along with the geometric optics retrieval algorithm which can't handle multipath well.
- The implementation of open-loop (start from SAC-C and COSMIC and thereafter) largely resolve the bias due to the tracking issue. In addition, the introduction of the holographic retrieval method (e.g., CT, FSI, PM) largely solve the multipath issue resulting from the "strong" refractivity gradient.
- Therefore, other than the "ducting", the refractivity biases can't be primarily attribute to the "strong" refractivity gradients.
- The authors need to rewrite the session to make it clear.

L71-72:
- The authors claimed that the ducting cause phase/SNR changes and thus introducing bending angle errors and additional refractivity errors. **The statement is NOT correct**. The primary reason of ducting inducted refractivity bias is due to the singularity problem in the Abel retrieval algorithm, which resulting in a non-unique inversion problem. Even

the receiver tracking is perfect, the bending angle do not have any errors, the refractivity biases will remain in the presence of ducting. The Abel retrieval always selects the minimum refractivity solution and thus large negative refractivity biases (Sokolovskiy 2001 etc.). Of course, the sharp gradient in the present of ducting could potentially adding complexity onto RO signal phase/amplitude and might introduce extra errors in RO bending angle and refractivity retrievals but is likely secondary.
- The paragraph needs to be rewritten.
- Similarly, in L153-154: the negatively biased refractivity is due to the singularity problem in Abel inversion, and thus the non-unique inversion problem.
- Same

L126-130
- Detailed description of the ERA5 data is missing. The spatial (grid size, vertical level #), temporal resolution of the data need to be described.
- In addition, how the collocation between the RO and ERA5 need to be clearly described, including the threshold for time and space differences.

L146-147:
- I don't think the statement is correct: "*However, in the presence of a large vertical gradient, refractivity is nonspherically symmetric, and noise appears because of multiple rays (Sokolovskiy 2010).*"
  o The large vertical gradient surely can introduce multipath but doesn't introduce the nonspherically symmetric atmosphere, unless there is horizontal inhomogeneity.
  o Rewritten sentence is needed.

L155-156:
- "Under certain conditions, extreme SR occurs, and the signal is trapped … called the atmospheric duct"
  o The authors seem to be confused of the difference between Super Refraction (SR) and ducting. A good explanation can be found in Lopez (2009). However, the earlier publications make the two terms interchangeable. For example, the SR introduced in Sokolovskiy 2003, Xie et al., 2006 and many others are all referring to the "ducting" condition, i.e., dN/dz < -157 N-unit/km.
  o There is no such thing of the "extreme SR" becoming ducting.
  o Also, Sokolovskiy (2003) shows that the RO signal entering the ducting layer from above will not be trapped. Only when the RO signal is from below the ducting layer could be trapped.

L221-222:    Figure 2b:
- Should overplot the percentage of profiles as a function of height in reference to the maximum number of profiles or the total number at 10 km.
- Although the authors claimed the number of profiles should still be sufficient for statistical evaluation, it is better to screen out the low sample regions (as indicated in Fig. 1), such as those grids with fewer than 30 profiles.

- Moreover, more stringent criteria should be explored, such as requiring the total number of profiles penetrating below 500 m above MSL within a grid to be more than at least 30 or more.

L315-318:
- It is not surprised to see weak relationship between ducting induced N-biases to the LSW, as the ducting induced N-bias is mainly caused by the Abel retrieval, but not too much due to the signal tracking or multipath (after implementation of open-loop tracking and radio-holographic retrieval method). The low temperature and low humidity (below 1.5 km) seem to be a good indicator for large N-bias over ducting region. However, such relationship could be serious fault over high latitude (not included in this study), for example, the polar region mostly with low temperature and moisture, but will not expect to see large N-biases but the opposite.
- The authors need to discuss the serious limitation of using T/Q for N-biases estimation away from subtropical oceans.
  - L331-332: *"This finding also confirms that the N-REFBs below 1.5 km are highly related to the thermodynamic conditions and that the TQ estimation successfully reflects the impact of the air-sea interaction on the RO refractivity."*
  - I don't think the "impact of air-sea interaction" has anything to do with the N-bias assessment here.
  - Also, such assessment of T/Q relation to N-bias WON'T be able to apply to higher latitude, and maybe even over land.
  - The limitation of use T/Q for global application need to be discussed.

L368:
- Figure 11 shows two selected regions (A & B). Why Area B shows the improvement in MVE than T/Q, which seems not consistent with Figure 10f, where large RMSE remained in Area B for MVE. Please explain.
- L620-625: Figure 11 and 12 can be combined into a three-panel plot for better presentation

Technical comments:

T & Q in the paper should be *tilted* as they are variables

L26: Change sentence:
- "… receive radio signals, which are emitted from GNSS transmitters and tend to bend"
  → "receive radio signals from GNSS transmitter, which tend to bend"

L30: "reflectivity" → refractivity; "reflect the changes → measure the change (to avoid the confusion

L52:   No need to add "atmospheric boundary layer (ABL)", should simply use PBL

L55:   Should clarify "the penetration rate of RO profile is limited to extremely moist conditions …". Is the RO sounding penetrating deeper in more moist condition?

L62-67: "strong refractivity gradient" has been used many times in this paragraph, But what does the "strong" means? The authors state that under "strong refractivity gradients, "In such conditions, the assumptions and approximation in the retrieval algorithm can result in large uncertainties in the RO data."  The refractivity gradient itself do not introduce the uncertainty. What assumptions and approximation are you referring to?

L89:   such as that → such that

L90:   Please clarify what the "reflected bending" means, does that mean "grazing" signal bending measurement?

L124: below the 1.5 km-height of sea level… → below the 1.5 km above the mean sea level…

L30:   Parathesis needed for the difference (REF_FS7 - REF_EC)

L181-182:
- Please explain why and how to "independent fitting is performed five times, by replacing the testing data with another 20% of the data".  So does the 20% testing data used in deriving the regression model? Will that defeat the purpose of using the "testing data" to verify the regression model?

L184:  5° x 3° → 5°longitude x 3° latitude?

L195:  product → processing

L199:  Equation (7): any explanation why only the normalized Q (specific humidity) was used in the higher order term, but not the temperature (T)? Should there be another $z_i^2$ term representing the T?

L209-213:      Equation (10):
- What is the meaning of the vector of ones "1". Why the
- What is $u_j$? explanation needed.

L224: Figure 3
- L588: "during the study period" should simply state the study period, e.g., from *** to **

L234:  "The increased LSW just above the boundary layers could be caused by common inversion layers in the troposphere of some oceans."
- As discussed in the major comment earlier, there is not any direct definition and calculation of PBL height is this study. In addition, regions especially over tropics might

not have clearly defined PBL due to convection. Would be more appropriate to simply using the height (e.g., 2 -3 km).
- Lager LSW due to "inversion layers", is this a pure guess or with certain supportive evidence? If so, the reference needs to be listed.

L244: "The LSW over ocean below 4 km increases faster over the ocean, and the second peak value at the PBL top is much larger"
- Again, the "PBL top" should be replaced with 2 km.

L254: "thermodynamic structure in the PBL" → "thermodynamic structure in the lower troposphere.

L267: "temperature and specific humidity averaged below 1.5 km"
- Discussion on the sensitive of the result due to the penetration issue should be added.

L270: "Instead, N-REFB appears at the PBL top"
- Again, please not use PBL top, but using the height, e.g., ~ 2km

L286: Figure 7
- It is not clear why ONLY the relationship between REFB and LSW over the southern hemisphere (austral summer) were discussed.
- The relationship over norther hemisphere should also be discussed. Same for Figure. 8
- L604: Can't see the orange open circle (Land data over 1.5 km). could consider using a slightly larger open circle to avoid overlapping by the solid circles.
  o Figure caption: "N-biases" should be replaced with REFB to be consistent with manuscript.
- What about the relationship in the northern hemisphere?
  o With the weaker N-bias during the evaluation period, will the relationship hold up? If not, please explain.
  o Note there are seasonal variations in N-bias, the northern hemisphere is expected to see higher N-biases in NH summer (JJA) season, will the authors expect to see clearer relationship in the NH as compare to the Fig. 7?

L293: N-biases → REFB

L294: Figure 8
- L297: The manuscript only discussed Fig. 8a, but not the Fig. 8b. Discussion of both are required, and the difference should be clearly stated.
- L609: No labels: (a) (b) in Fig. 8. Also Figure caption should be self-explanatory. The manuscript also did not clearly tell the difference. Caption should clearly indicate the difference between the two panels, one for ocean and the other over land, right?
- The relationship is hard to be seen clearly.
- Similar to Fig. 7, the relationship over norther hemisphere should also be presented and discussed.

L318: rewrite "inversion layers on top of the surface cold atmosphere"? Do you mean on top of cool sea surface?

L351: Figure 10
- "Almost all the large RMSEs in the LSW or TQ estimation are removed by the MVE method (Fig. 10c and 10f)"
    o It is hard to see the significant improvement from MVE results (10 e,f) as compared to TQ-based approach (10 c,d). Actually, a slight degradation can be seen.
    o Need to explain why the training data set show the large improvement of T/Q over ducting region is SE Pacific/Atlantic in Fig. 10-c,e (e.g., SE Pacific [30S,90W]), but was not the case for the testing data set (Fig. 10-d,f)
    o L355:

L366: EQ < Lat < 10°N → 0° < Lat <10°N

L383: 3 N → 3 N-unit

L386: Please remove the PBL in "positive bias above the PBL altitudes of 3 to 5 km."

L423: "… temperature and moisture from the ERA5 reanalysis may have their own bias

L612: Box A & B should include the [lat, lon] information.

---

## Author Comment (AC1)

Dear Dr. Anthes,

We deeply appreciate your careful reading, editing and insightful comments and suggestions, which have greatly improved our manuscript. We have revised our manuscript significantly to address your comments and suggestions. Please see our point-by-point response (in blue) to your comments/suggestions as follows.

**Major Comments:**

1. The potential use of this statistical technique to estimate the likelihood and magnitude of refractivity biases in individual RO observations for data assimilation (DA) could be mentioned after line 108. However, most global models assimilate bending angles (BA) rather than refractivity. Did the authors try their statistical model to estimate BA biases? If so, they could summarize what they found. If not, this could be a topic for further study.

   Thank you very much for this important comment. We now clarify the motivation of this study in this study.

   - We have mentioned the potential use of the statistical technique proposed in this study in line 106-110 to address reviewer's comment.

     Our study focuses on the refractivity bias (REFB) in the lower troposphere for two reasons. First, we would like to use the estimation model to understand the characteristics of REFB and how they link to LSW and thermodynamic condition in the lower atmosphere, particularly in the planetary boundary layer. The bias estimation can be used to calibrate RO refractivity, which can be applied to improving the products of temperature and moisture profiles retrieved from the refractivity in the moist lower troposphere or estimating precipitable water vapor (Yeh et al. 2024). Second, the estimated REFB can be used for data assimilation purposes. With the DA systems that assimilate the RO-REF profiles, it is expected that the RO data in the lower atmosphere can be better exploited by using the bias estimation as a QC flag or assimilating the calibrated REF profiles. Although most global models assimilate the bending angles, the data has large uncertainties in the moist lower troposphere or is unavailable below the ducting layer. It is unclear whether the assimilating bending angle can positively impact low-level moisture accuracy. If the RO refractivity can be calibrated, they can likely be used in these conditions!

Here, we show an example of the RO profile over the ocean with moist PBL. The ECMWF 12-h forecast (echPrf) suggests the potential existence of ducting. The RO refractivity seems negatively biased below the ducting layer (black dashed lines), while the RO bending angle has significant variations in the low levels and quickly decreases to a very small value near the surface.

[Figure]

**Figure A An example of a RO profile with the existence of ducting over South China Sea. The atmPrf and echPrf of (a) bending angle and (b) refractivity. (c) Vertical distribution of the temperature and due-point temperature at this location from the ECMWF 12-h forecast.**

2.  A key part of this paper is the regression model for REFB vs. LSW or T and q. It is not clear how the two regression equations for LSW and for T and q were obtained. Why is Eq. (5) a quadratic in LSW and not some other relationship (e.g. linear, cubic, or higher order)? Why is Eq. (7) quadratic in Q (normalized specific humidity) plus the product of normalized Q and T? What other polynomials were tested? Presumably some of the process in selecting the optimum polynomial is described in lines 181-188, but additional detail would be useful.

We apologize for the unclear description of our methodology in the previous version of the manuscript. Different combinations of the order of the predictors are evaluated to define the optimal equation (polynomial) to best represent the real REFB. This section (section 2.2) has been revised significantly to clarify and address your comments. In the following, we explain in more details about how to find out the best formula of each participated variable and their associated coefficients.

The feature engineering and model selection tools in Scikit-learn (Python 3.1) are used to evaluate the fitting performance with the polynomial regression model. The evaluation is conducted for the 'degree' parameter ranging from 1 to 6 with the metrics of R-squared ($R^2$) and mean squared error (MSE).

The following figure shows the $R^2$ of the LSW-based polynomial with different degree (order), and the corresponding computational time. Although a higher R² is obtained with the higher degree of

polynomial, the computational cost is increased. The figure shows that R² significantly increases from a linear to quadratic form of the polynomial and becomes saturated. In comparison, the computational cost increases linearly as the degree increases. This suggests that the evident gain of the performance is obtained from the linear to quadratic form and saturates for higher order. To avoid overfitting and consider the computational time, the quadratic form was chosen for the LSW-based REFB model.

[Figure]

**Figure B The performance and computation time with respect to different degree of polynomial regression using LSW as the predictor**

For the TQ-based model, there are two variables ($T$ and $Q$) used in the regression model and the product term ($TQ$) is included to consider the joined effect (interaction) between these two variables. The following table lists $R^2$ and MSE of polynomials with different variable combinations. Results show that the formulas including $Q^2$, $Q$ and $TQ$ terms gives the best fitting performance. In particular, including the quadratic term of moisture is essential. The R-squared value increases from 0.535 with the $Q$ and $TQ$ terms to 0.732 with the $Q^2$ term. However, $TQ^2$ or $T^2Q$ only mildly increases $R^2$ but degrades MSE (Table A). This suggests that the high order interactive terms do not add significant benefit to the fitting. Therefore, we choose the formula with $Q^2$, $Q$ and $TQ$ terms.

We have briefly included the discussion in line 184-187, 191-192.

**Table A: $R^2$ and MSE of polynomials with different variable combinations**

| Equation forms | Order of variables: 1 | | | | | Order of variables: 2 | | | | | | | |
|---|---|---|---|---|---|---|---|---|---|---|---|---|---|
| | T and Q | TQ and const | Q, TQ | $Q^2$, Q | $Q^2$, Q, TQ | $Q^2$, Q, TQ² + TQ | $Q^2$, T²Q, TQ² ,TQ, Q | $Q^2$,T², TQ, TQ², TQ, Q, T | $Q^2$,T², T²Q, TQ ,T | T²,T²Q, TQ ,T | T²,T, T²Q | T²,T | T²,Q² |
| $R^2$ | 0.399 | 0.421 | 0.535 | 0.481 | 0.732 | 0.734 | 0.734 | 0.689 | 0.695 | 0.245 | 0.115 | 0.111 | 0.211 |
| MSE | 77.267 | 72.115 | 37.044 | 42.886 | 26.610 | 26.611 | 26.611 | 36.614 | 34.205 | 94.336 | 101.591 | 109.460 | 98.629 |

3. In Section 2.3 the authors say that they derived the statistical models using the data for five different subsets of the data and chose the ones with the best fits as their model. I am not an expert in statistics, but why not use the entire sample of data for their statistical model? The "best" model for one subset may not be the "best" model overall? And if more subsets (e.g. 10) were chosen, a different "best" model would be obtained. This issue should be discussed.

Splitting the data into training and testing subsets is commonly adopted for constructing a statistical model. The training data is used to build the regression model. The testing data, independent of the training data, is used to evaluate the performance of the regression model. The fitting performance may be very good with the training data but poor with the testing data (i.e. an overfitting case). Therefore, the testing data is required to avoid overfitting and ensure the robustness of the derived model.

Theoretically, we need sufficient (randomly generated) training data to capture the general behaviors of the data. Given that the sample size is limited, we try to find the most representative regression model by repeating the training procedure with different data portions (80%) and evaluating the fitting performance with the rest of the data (20%). The regression model that can best fit the testing data is selected. With 20% data as the testing data, the replacement of the testing data is repeated five times so that the regression model is eventually applied to the entire data set.

The applications of the regression models are conducted in two parts.

(1) All profile data (80% for training and 20% for testing) is used to determine the order of the LSW-based regression model and the optimal combination of the multi-variable (T and Q) regression model (Given two variables $y$ and $z$, there are different combinations of order and interaction terms as $\sum_{m=0}^{m=M} \sum_{l=0}^{l=L} b_{m,l} y^m z^l$, where $m$ and $l$ are the order of variable $y$ and $z$, respectively, and $b_{m,l}$ is the regression coefficient).

(2) The forms of the polynomial regression are then applied regionally. From 45°S to 45°N, we define 72 x30 boxes and each box is 5° longitude x 3° latitude degree. The boxes are defined by considering the number of available RO profiles below 1.5km should be sufficient for regression testing. With the 3 months of data used in our study, choosing testing data lower than 20% of the total sample results in a very coarse resolution of the boxes. Choosing any number larger than 20% would sacrifice the amount of data that can train a reliable regression model. In each box, the regression fitting is also repeated five times to derive the optimal regression coefficients.

The discussion above has been included in the section of methodology (section 2.2).

**Specific comments:**

1. Line 123: Fig. 1 is labeled profile density, but it is actually profile counts. The label should be changed.

   We have corrected this, thank you!

2. Line 128: The issue with possible biases in the reference (ERA5) should be mentioned here.

   Thank you for pointing this issue. We have addressed this comment accordingly in line 136.

   "Nevertheless, it is possible that ERA5 may carry its own biases, which will not be discussed in this study"

3. Sometimes N-REFB is used and other times REFB is used. Please be consistent. Since you are only discussing refractivity biases, I suggest using just REFB. When referring specifically the negative biases, I suggest saying "negative REFB."

   We apologize for the inconsistency. We now only use REFB and replace N-REFB to negative REFB as the reviewer suggested.

4. 5 caption: I suggest emphasizing that the REFB, LSW, q and T in Fig. 5 are all averages over the lowest 1.5 km MSL. Add to the caption: "The values of REFB, LSW, specific humidity and temperature are averages over the lowest 1.5 km MSL of the atmosphere." And in line 251 write "the averaged value of REFB below 1.5 km"

   Thank you for your great suggestion. This is emphasized in the caption of figure 5 and wherever it is necessary.

5. 6 caption: "Vertical cross section of refractivity bias over the ocean as a function of height and average values over the lowest 1.5 km of (a) LSW/2, (c) specific humidity and (e) temperature over land..........

   Thank you for your suggestion. We have corrected the caption of Figure 6.

6. Lines 131-173 (Section 2.2)

   This section contains some incorrect or misleading statements and is unnecessary for this paper. For example, in Lines 144-146: "Normal" is not well defined; nonspherically symmetric conditions are common. Spherically symmetric means no horizontal variation of refractivity on a constant level surface, either small-scale turbulent variations in T and q or larger-scale horizontal gradients

of T and q. In line 146, a large vertical gradient of refractivity does not necessarily imply nonspherical symmetry, which depends on horizontal variations of N not vertical gradients. I suggest a much shorter simplified summary that refers to more complete discussions of the causes of negative refractivity biases. Here is an example:

"Negative refractivity biases can arise in the atmosphere from multiple causes, as summarized by Feng et al. (2020) and Wang et al. (2020). A common cause (but not the only one) of negative biases in the lower troposphere is ducting or superrefraction (Sokolovskiy 2003; Ao et al. 2003). When the vertical gradient of refractivity $\partial N/\partial z$ exceeds a critical value of -157 N units per km, ducting occurs and rays are trapped inside the ducting layer. This leads to a negative bias in N and there are an infinite set of bending angle profiles that correspond to the observed refractivity profile."

We deeply appreciate your corrections and clarification for the nonspherical symmetry and vertical gradient of refractivity. Since this section is greatly reduced, we merged the main sentences to the introduction to review the causes of negative refractivity bias (line 59-75).

7. Line 175: Why not use LSW as a predictor instead of LSW/2? The correlations should be the same.

The LSW is the standard deviation of bending angle, which defines the data uncertainty. In Liu et al. (2018), LSW/2 is used as a predictor and the rescaling factor is to represent the standard deviation of a Gaussian distribution. This is now clarified in line 147-148.

8. Line 194—what 1D-Var algorithm was used? The original CDAAC wetPrf or the new CDAAC wetPf2 (Wee et al. 2023)? Or some other one?

The wet product we used in the study is the 1D-var wetPrf2. The wetPf2 refractivity is derived from the bending angle profile of atmPrf and through a variational regularization of Abel transform (Wee, 2018), instead of the traditional Abel inversion. We have clarified that the T and Q obtained from the 1D-Var analysis of the RO wet products in line180.

Wee, T.-K.: A variational regularization of Abel transform for GPS radio occultation, Atmos. Meas. Tech., 11, 1947–1969, https://doi.org/10.5194/amt-11-1947-2018, 2018.

9. Lines 255-257: there is no direct relationship between large vertical gradients of N and nonspherical symmetry, which is caused by horizontal variations in N. Large horizontal variations in N and corresponding large LSW may occur with small vertical gradients of N. I suggest deleting the two sentences in lines 255-257; the previous sentence is sufficient.

We apologize for the misleading statement. Thank you very much for your correction. These sentences are deleted in the revised manuscript.

10. Which is land and which is ocean in Fig. 8? The caption should provide more details, i.e. explain the dots, explain the surface (is it a fit to the dots?)

Thank you for your comments. For better illustration, we have re-plotted Figure 8 to be function of normalized $Q$ and $TQ$ over ocean (Fig. 8a) and land (Fig. 8b) in Southern Hemisphere. We also included Figs. 8c and 8d to show the same plot for Northern Hemisphere. Details about this figure are provided in the caption.

11. Lines 311-312 and Fig. 9: Why aren't the "Real REFB" the same for the training and testing data? Is this a sampling issue?

The training and testing data are 80% and 20% of the total data, respectively. The real REFB in Figs. 9a and 9b are calculated with the training and testing data, respectively. Therefore, it is expected that the pattern should be very similar but have some differences.

---

## Author Comment (AC2)

Dear Reviewer,

We sincerely appreciate your careful reading and insightful comments/suggestions, which have greatly improved our manuscript. We have significantly revised our manuscript to address your suggestions. We have made several major changes to our manuscript.

(1) We removed section 2.2 and reviewed the causes of negative bias in the introduction.

(2) We included a new section to discuss sensitivity experiments suggested by the reviewers.

(3) Several figures are re-plotted with a new color bar for better illustration.

(4) We included new figures (or subplots) to address the sampling issues raised by the reviewer.

The manuscript has also been significantly revised following the suggestions and comments of another reviewer, Dr. Richard Anthes. Please see our point-by-point response (in blue) to your comments/suggestions as follows.

**Major Comments:**

1. The planetary boundary layer has been raised in numerous places in the paper, including the paper title. However, there are no clear definition of PBL, instead of a fixed height of 1.5 km as the indication for PBL. Note that the PBL height has various definition and can change significantly from a few hundred meters up to 3-4 km. It might be too much simplification for use one fixed height of 1.5 km to indicate the PBL.

   We understand the reviewer's concern about using a fixed height of 1.5km to represent the PBL height. To address the reviewer's comment, we have replaced PBL with "deep troposphere" in the paper title and specific the height instead of using top of PBL. But to justify the choice of the 1.5 km height used in many figures, we also emphasize that the 1.5 km height is the global mean PBL height calculated from the COSMIC refractivity (Xie 2014) (line 253-254).

   Xie, F, Visiting Scientist Report 21: Investigation of methods for the determination of the PBL height from RO observations using ECMWF reanalysis data, SAF/ROM/DMI/REP/VS21/001, 2014.

2. The authors need to clearly stating that the Local spectral width, Temperature (T) and specific humidity (Q) used in this paper are the "average" below 1.5 km?

   Thank you for your suggestion. We now emphasize that the LSW, T and Q that is used to construct Figures 7 to Figures 9 are the average below 1.5km. Please see the figure captions.

   L267: Temperate and specific humidity averaged below 1.5 km for land and ocean (Figure 6)

   - Do the authors considering the regional difference in penetration depth of RO profiles (Fig. 1)?

Yes, for Figs. 6c-6f, the REFB (refractivity bias) is grouped according to the temperature and specific humidity averaged below 1.5km for land and ocean. The regional difference in penetration depth of RO profiles is not considered in Fig. 6.

- How to deal with the potential negative bias in T/Q resulting from the early termination of the RO profiles, i.e., missing the lowest part of profile below 1.5 km where temperature and moisture values are likely larger closer to the surface.

When constructing the $TQ$-based regression model, $T$ and $Q$ are taken from the 1D-Var analysis product of the wet products and REFB, originally calculated from atmPrf, will be interpolated to the same levels of wetPf2. Only were the data below 1.5 km used. In other words, if the profile of the wet product terminates early, there will be no $T$ and $Q$ available. We now clarified this in line 181-183.

- Do the authors process all RO data or only the ones penetrating certain threshold (1.5 km??)? Sensitivity test might be needed, for example, regenerate Fig. 5 by requiring RO profiles to penetrate below 500m above mean-sea-level. Or simply calculate the average between 500m and 1.5 km.

Yes, we used only the RO profiles penetrating 1.5 km above MSL.

Following the reviewer's request, we have conducted sensitivity tests with different criteria to select the samples. Please see our discussion in the new subsection 4.2.

The follow figure (Figure A) compares REFB, LSW, $T$ and $Q$ averaged below 1.5 km (Figure 5 in the manuscript) with the criterion that at least 30 profiles penetrate below 500 m in each box (defined as the C2 criterion). With a stringent criterion, there are insufficient samples in mid-latitude, Africa and South America and partial ducting regions. For boxes with sufficient samples, the patterns of REFB, LSW, $T$ and $Q$ are very similar to the ones with an eased standard criterion (at least 10 profiles in each box) (and the ones using RO data between 0.5 to 1.5 km), but the amplitudes are generally higher. We now include sentences to emphasize that results are dominated by the data in between sentences 0.5 and 1.5 km. The REFB estimation using the C2 criterion (right column of Figure B) still show good ability to capture the characteristics of real REFB, even in the regions (central and northwestern Pacific) that the real REFBs are somewhat different between the C2 and standard criteria. This good performance is attributed to the fact that the region-dependent regression models can adapt to the changes in the training data in boxes.

Both Figure A and Figure B are now included in the revised manuscript as the revised Figure 5 and Figure 12, respectively. The relevant discussion is included in the new section 4.2 of sensitivity experiments.

[Figure]

**Figure A: Horizontal distribution of (a) REFB (*N units*), (c) LSW (%), (e) specific humidity (g kg⁻¹), and (g) temperature (°C) averages over the lowest 1.5 km MSL of the atmosphere. (b), (d), (f) and (h) are the same as (a), (c), (e) and (g), except that they are calculated with a stringent criterion requiring at least 30 profiles penetrating below 0.5 km in each box.**

[Figure]

**Figure B (a) Real, (c) LSW-based, (e) *TQ*-based, and (g) MVE REFB with different criteria for selecting the samples. (i) is the horizontal distribution of the total number of RO profiles with the criterion that at least 30 profiles penetrate below 1.5 km. (b), (d), (f), (h), and (j) are the same as (a), (c), (e), (g) and (i), except with the criterion that at least 30 profiles penetrate below 0.5 km.**

3.  I am not totally convinced that the temperature and specific humidity patten correlated to the refractivity bias.

    The seemingly high correlation between temperature/specific-humidity (T/Q) and the refractivity biases over the ducting regions is not a surprise, as those regions are mostly over the ocean upwelling region with cool sea surface temperature and relatively low moisture in PBL (below 1-2 km). However, the T/Q relationship to the REFB might not hold over high latitude.

    We understand the reviewer's concern and appreciate the insightful comment about the potential limitation of the *TQ*-based REFB. As shown in Fig. 10 (the old Fig. 9), the *TQ* estimator not only well estimates the amplitude of REFB in the ducting regions but also in the non-ducting areas like north western Pacific, western Atlantic and equatorial region.

    However, given that the Formosat7/COSMIC-II RO profiles are mostly located in tropic to subtropic region. We have limited data to evaluate whether how the *TQ* estimator performs over high latitude.

    We address the reviewer's comment from two aspects:

    (1) Figure 8 suggests that the estimated REFB becomes slightly positive for the conditions with very low

temperature and moisture (*TQ* and *Q* close to zero), i.e. high latitude condition. Qualitatively, this agrees with the characteristics of REFB over Bering Ocean shown in Fig. 4a of Feng et al. 2020). Since this area is not covered by the RO data used in this study, it is difficult to justify whether the regional-dependent TQ estimator is applicable to estimate REFB in the polar or high-latitude regions. Therefore, it is still an open question and may need to be evaluate with more RO data from other satellites. We included the discussion for Fig. 8 (line 302-307) and in the summary (line 465-467).

(2) In subsection 4.3, we now include another box C located offshore of higher-latitude north America. The *TQ* estimator captures the general pattern of the vertical distribution of REFB but the amplitude is smaller than the real REFB. Nevertheless, the *TQ*-based REFB is much better represented compared to one from the LSW estimator.

4. L67-70:

- The authors give the impression that the "strong refractivity gradient" are the primary problem or RO measurement uncertainty. The authors claimed that the strong vertical refractivity gradient will introduce multipath and resulting complex bending angle and complexity of RO uncertainty. Also the L70: "strong refractivity" (I assume it should be strong refractivity gradient) causes a negative refractivity bias (N-REFB) (Rocken et al., 1997).

- However, the negative bias in lower troposphere in earlier generation RO missions (e.g., Rocken et al., 1997) were mainly due to the close-loop tracking issue in early generation of RO receivers, along with the geometric optics retrieval algorithm which can't handle multipath well.

- The implementation of open-loop (start from SAC-C and COSMIC and thereafter) largely resolve the bias due to the tracking issue. In addition, the introduction of the holographic retrieval method (e.g., CT, FSI, PM) largely solve the multipath issue resulting from the "strong" refractivity gradient.

- Therefore, other than the "ducting", the refractivity biases can't be primarily attribute to the "strong" refractivity gradients.

- The authors need to rewrite the session to make it clear.

  Thank you very much for the corrections. This paragraph (line 59-70) is significantly revised with the following sentences to address reviewers' comments and suggestions.

  *"The implementation of open-loop tracking (Sokolovskiy, 2001) and the use of the holographic retrieval method largely reduce the negative refractivity bias (REFB) in lower troposphere in earlier generation RO missions. The "radioholographic" methods such as the canonical transform (CT) method (Gorbunov, 2001, 2002), Full Spectrum Inversion (FSI) (Jensen et al, 2002) and Phase matching (PM) (Jensen et al, 2004) largely solve the multipath issue resulting from the "strong" refractivity gradient"*.

  We now only emphasize that under strong vertical refractivity gradients, ducting is the main source that cause the negative bias of refractivity due to the Abel inversion (line 68-70).

5. L71-72:
   - The authors claimed that the ducting cause phase/SNR changes and thus introducing bending angle errors and additional refractivity errors. The statement is NOT correct. The primary reason of ducting inducted refractivity bias is due to the singularity problem in the Abel retrieval algorithm, which resulting in a non-unique inversion problem. Even the receiver tracking is perfect, the bending angle do not have any errors, the refractivity biases will remain in the presence of ducting. The Abel retrieval always selects the minimum refractivity solution and thus large negative refractivity biases (Sokolovskiy 2001 etc.). Of course, the sharp gradient in the present of ducting could potentially adding complexity onto RO signal phase/amplitude and might introduce extra errors in RO bending angle and refractivity retrievals but is likely secondary.
   - The paragraph needs to be rewritten.
   - Similarly, in L153-154: the negatively biased refractivity is due to the singularity problem in Abel inversion, and thus the non-unique inversion problem.
   - Same

Thank you for your careful reading and suggestions. We have addressed your comments and rewritten this paragraph to summarize the causes of negative refractivity bias (line 59-75).

In particularly, we emphasize that in the presence of ducting, the singularity problem in the Abel transforms leads to a non-unique inversion problem. Thus, the Abel inversion results in a negatively bias refractivity below the ducting layers (Sokolovskiy, 2003).

Please see the new paragraphs in line 69-70.

6. L126-130
   - Detailed description of the ERA5 data is missing. The spatial (grid size, vertical level #), temporal resolution of the data need to be described.
   - In addition, how the collocation between the RO and ERA5 need to be clearly described, including the threshold for time and space differences.

Thank you for your suggestions. We have included the detailed description in line 129-134:

"*For comparison, we used the European Centre for Medium-Range Weather Forecasts (ECMWF) atmospheric reanalysis (ERA5, https://www.ecmwf.int/en/forecasts/access-forecasts/access-archive-datasets) as the reference RO profiles. The hourly ERA5 reanalysis in the study period has a horizontal resolution of 0.25 x 0.25 deg grid and 37 pressure levels ranging from 1hPa to 1000hPa. The variable geopotential, temperature and specific humidity are selected. Since the time of the RO data is precise in minutes, we rounded RO time to the nearest hour. The ERA5 profiles are derived by interpolating the*

*reanalysis horizontally and vertically to the location and vertical levels of the RO profiles.*"

7. L146-147:

I don't think the statement is correct: "However, in the presence of a large vertical gradient, refractivity is nonspherically symmetric, and noise appears because of multiple rays (Sokolovskiy 2010)."
The large vertical gradient surely can introduce multipath but doesn't introduce the nonspherically symmetric atmosphere, unless there is horizontal inhomogeneity.
Rewritten sentence is needed.

*We apologize for the confusing sentences and appreciate all the important references. Following Dr. Anthes's suggestion, most of the discussion in this section are removed. Therefore, we merged the main sentences to the introduction (line 59-75) to briefly review the cause of negative refractivity.*

8. L155-156:

- "Under certain conditions, extreme SR occurs, and the signal is trapped … called the atmospheric duct"
- The authors seem to be confused of the difference between Super Refraction (SR) and ducting. A good explanation can be found in Lopez (2009). However, the earlier publications make the two terms interchangeable. For example, the SR introduced in Sokolovskiy 2003, Xie et al., 2006 and many others are all referring to the "ducting" condition, i.e., dN/dz < -157 N-unit/km.
- There is no such thing of the "extreme SR" becoming ducting.
- Also, Sokolovskiy (2003) shows that the RO signal entering the ducting layer from above will not be trapped. Only when the RO signal is from below the ducting layer could be trapped.

*We apologize for the confusing sentences and appreciate all the important references. Following Dr. Anthes's suggestion, most of the discussion in this section are removed. Therefore, we merged the main sentences to the introduction (line 59-75) to briefly review the cause of negative refractivity.*

9. L221-222: Figure 2b:

- Should overplot the percentage of profiles as a function of height in reference to the maximum number of profiles or the total number at 10 km.

    *Thank you for this great suggestion! We now include Fig. 2c to show the percentage of profiles as a function of height in reference to the maximum number of profiles or the total number at 10 km.*

- Although the authors claimed the number of profiles should still be sufficient for statistical evaluation, it is better to screen out the low sample regions (as indicated in Fig. 1), such as those grids with fewer than 30 profiles.
- Moreover, more stringent criteria should be explored, such as requiring the total number of profiles

penetrating below 500 m above MSL within a grid to be more than at least 30 or more.

Following your suggestion, we have conducted REFB estimation and evaluation for boxes with more than 30 profiles penetrating below 1.5 and 0.5km, respectively.

Figure B shows the REFB estimation with different criteria for the penetration rate. The estimators are obtained when there are at least 30 profiles whose minimum level is smaller than 1.5 or 0.5 km, respectively. The experiments are referred to as CT1 and CT2. As the criterion becomes more stringent, more samples in the tropics are rejected and insufficient samples are available in the core region of the ducting regions and areas with latitudes higher than 30 degrees. Nevertheless, the real REFB in Fig. B is very similar to those in Fig. 10 (the old Fig. 9) using an eased criterion on sample number. This similarity reflects that the REFB is dominated by the values between 0.5 and 1.5 km. Furthermore, the LSW-based REFB with strict criteria also captures the general pattern of real REFB, while the TQ-based REFB captures the large negative REFB in the ducting regions well. The estimated REFB from the three methods in Fig. B is also similar to those in Fig. 10. Even with stringent criteria, LSW, T, and Q above 0.5 km carry essential information about REFB in tropic-to-subtropic regions.

Figure B is now included in the revised manuscript as Figure 12 and the relevant discussion is included in the new section 4.2 of sensitivity experiments.

10. L315-318:

- It is not surprised to see weak relationship between ducting induced N-biases to the LSW, as the ducting induced N-bias is mainly caused by the Abel retrieval, but not too much due to the signal tracking or multipath (after implementation of open-loop tracking and radio-holographic retrieval method). The low temperature and low humidity (below 1.5 km) seem to be a good indicator for large N-bias over ducting region. However, such relationship could be serious fault over high latitude (not included in this study), for example, the polar region mostly with low temperature and moisture, but will not expect to see large N-biases but the opposite.

- The authors need to discuss the serious limitation of using T/Q for N-biases estimation away from subtropical oceans.

Thank you for your suggestions.

Given that LSW quantifies the uncertainties of bending angle, LSW is expected to carry the REFB inherited from bending angle. Feng et al. (2020) pointed out that there are non-ducting related biases exist in the RO data. Error associated with low SNR in the complex moist lower troposphere may cause negative biases in bending angles and refractivity. Another potential source is the propagation of radio

waves in a medium with random refractivity irregularities can also cause biases (Gorbunov et al. 2015). (line 71-75, line 176-179). Despite that the LSW-based REFB underestimates the bias amplitude in the ducting regions, it captures the general patten of REFB, as reflected with a high correlation coefficient between the mean LSW and REFB below 1.5 km. Such relationship is also mentioned in Liu et al. (2018). We have included the discussion to address your comment (line 146-147). When focusing on the REFB below 0.5 km (Fig. C), the LSW-based estimator has a great performance for capture the general pattern and amplitude of REFB. Fig. C is included in the new Fig. 13.

[Figure]

**Figure C Horizontal distribution of (a) real and (b) LSW-based REFB estimation below 1.5 km**

The predictors $T$ and $Q$ are expected to represents the biases related to the characteristics of the thermodynamic structure in deep troposphere, such as the ducting regions with a cool-dry PBL over subtropical oceans. However, it should be noted that the current TQ estimator is obtained based on the Formosat-7/COSMIC-II data, which mostly distributed in the tropic to subtropic regions. Nevertheless, Figure 8 suggests that the estimated REFB becomes slightly positive for the conditions with very low temperature and moisture ($TQ$ and $Q$ close to zero). Qualitatively, this agrees with the characteristics of REFB over Bering Ocean (Feng et al. 2020). However, whether the regional-dependent TQ estimator is adequately applied to estimate REFB in the polar or high-latitude regions is still an open question and may need to be evaluate with more RO data from other satellites.

We have included the relevant discussion for Fig. 8 (line 302-307) and in the summary section (line 465-467).

11. L331-332: "This finding also confirms that the N-REFBs below 1.5 km are highly related to the thermodynamic conditions and that the TQ estimation successfully reflects the impact of the air-sea interaction on the RO refractivity."

I don't think the "impact of air-sea interaction" has anything to do with the N- bias assessment here.

This sentence is now revised from the revised manuscript.

Also, such assessment of T/Q relation to N-bias WON'T be able to apply to higher latitude, and maybe even over land.

The limitation of use T/Q for global application need to be discussed.

We addressed the potential limitation of using the *TQ* estimator in the discussion for Fig. 8 (line 302-307) and in the summary (line 465-467).

12. L368:

Figure 11 shows two selected regions (A & B). Why Area B shows the improvement in MVE than T/Q, which seems not consistent with Figure 10f, where large RMSE remained in Area B for MVE. Please explain.

It should be noted that Fig. 11 (the old Fig. 10) is calculated based on the difference the real REFB and estimated REFB of each profile "averaged" below 1.5 km, where Fig. 14 (the old Fig. 11) groups the profiles with an interval of 500m. Therefore, the overestimation REFB below 1km with the TQ-based estimator will not be reflected with the average data used to construct Fig. 11. We have included the relevant explanation when discussing Fig. 11 (line 413-417).

L620-625: Figure 11 and 12 can be combined into a three-panel plot for better presentation.

Thank you for your suggestion. Figures 11 and 12 are combined into a multi-panel plot. Please refereed to the new Figure 14.

**Specific comments:**

Technical comments:

T & Q in the paper should be ***tilted*** as they are variables

Thank you for your suggestion. T and Q are tilted in the revised paper.

L26:    Change sentence:
   -    "… receive radio signals, which are emitted from GNSS transmitters and tend to bend"
        → "receive radio signals from GNSS transmitter, which tend to bend"
        This sentence (line 26) is corrected accordingly.

L30:    "reflectivity" → refractivity; "reflect the changes → measure the change (to avoid the confusion
        Thank you for the suggestion. This sentence (line 30) is corrected and modified. "reflect the changes" is now modified to "measure the vertical gradients" following Dr. R. Anthes's suggestion.

L52:    No need to add "atmospheric boundary layer (ABL)", should simply use PBL
        This sentence (line 52) is corrected accordingly. And only PBL is used throughout the revised manuscript.

L55: Should clarify "the penetration rate of RO profile is limited to extremely moist conditions …". Is the RO sounding penetrating deeper in more moist condition?

In Anthes et al. (2022), they did not conduct statistical comparison for the lowest level of the RO penetration. But, they did show that the most of the RO sounding have a deep penetration below 1 km in very moist condition. This sentence (line 55) has been modified according to Dr. Anthes's suggestion "*Anthes et al. (2022) noted that the penetration rate of RO profiles is high even under extremely moist conditions and near tropical cyclones.*"

L62-67: "strong refractivity gradient" has been used many times in this paragraph, But what does the "strong" means? The authors state that under "strong refractivity gradients, "In such conditions, the assumptions and approximation in the retrieval algorithm can result in large uncertainties in the RO data." The refractivity gradient itself do not introduce the uncertainty. What assumptions and approximation are you referring to?

Following Lopez (2009), ducting occurs with conditions of strong vertical refractivity exceeding -157 N-units km$^{-1}$ (line 68).

This paragraph is rewritten (line 59-75) and the sentence "In such condition, the assumption and approximation..." is removed from this paragraph for fluency.

L89: such as that → such that

Corrected. Thank you.

L90: Please clarify what the "reflected bending" means, does that mean "grazing" signal bending measurement?

Yes, it is the grazing signal of bending measurement. This is clarified in the same sentence (line 88).

L124: below the 1.5 km-height of sea level… → below the 1.5 km above the mean sea level…

Corrected (line 127). Thank you.

L130: Parathesis needed for the difference (REF_FS7 - REF_EC)

The old equation (1) has been removed following Dr. Anthes's suggestion.

L181-182:

- Please explain why and how to "independent fitting is performed five times, by replacing the

testing data with another 20% of the data". So does the 20% testing data used in deriving the regression model? Will that defeat the purpose of using the "testingdata" to verify the regression model?

Since the testing data is chosen as 20% of the total data, the independent fitting is performed five times with different 20% of the data so that the testing data from five experiments eventually covers the whole data set. It should be noted that 80% of the data is used to derive the regression model (i.e. train the model). We then used the rest of 20% of the data to evaluate the performance of the model to avoid the potential over-fitting issue with the training data.

We have clarified the fitting procedure in line 157-171.

L184: 5° x 3° → 5°longitude x 3° latitude?

Corrected (line 164). Thank you.

L195: product → processing

The word "product" has been replaced by 1D-Var analysis (line 181), as suggested by Dr. Anthes.

**L199**: Equation (7): any explanation why only the normalized Q (specific humidity) was used in the higher order term, but not the temperature (T)? Should there be another $z_i^2$ term representing the T?

For the TQ-based model, there are two variables ($T$ and $Q$) used in the regression model and the product term ($TQ$) is included to consider the joined effect (interaction) between these two variables. The following table lists $R^2$ and MSE of polynomials with different variable combinations. Results show that the formulas including $Q^2$, $Q$ and $TQ$ terms gives the best fitting performance. In particular, including the quadratic term of moisture is essential. The R-squared value increases from 0.535 with the $Q$ and $TQ$ terms to 0.732 with the $Q^2$ term. However, $TQ^2$ or $T^2Q$ only mildly increases $R^2$ but degrades MSE (Table A). This suggests that the high order interactive terms do not add significant benefit to the fitting. Therefore, we choose the formula with $Q^2$, $Q$ and $TQ$ terms. This is clarified in line 186-189 and 191-192.

**Table A: $R^2$ and MSE of polynomials with different variable combinations**

| Equation forms | Order of variables: 1 | | | | | | Order of variables: 2 | | | | | | |
|---|---|---|---|---|---|---|---|---|---|---|---|---|---|
| | T and Q | TQ and const | Q, TQ | $Q^2$, Q | $Q^2$, Q, TQ | $Q^2$, Q, TQ² + TQ | $Q^2$, T²Q, TQ² ,TQ, Q | $Q^2$,T², T²Q, TQ², TQ, Q, T | $Q^2$,T², T²Q, TQ ,T | T²,T²Q, TQ,T | T²,T, T²Q | T²,T | T²,Q² |
| $R^2$ | 0.399 | 0.421 | 0.535 | 0.481 | 0.732 | 0.734 | 0.734 | 0.689 | 0.695 | 0.245 | 0.115 | 0.111 | 0.211 |
| MSE | 77.267 | 72.115 | 37.044 | 42.886 | 26.610 | 26.611 | 26.611 | 36.614 | 34.205 | 94.336 | 101.591 | 109.460 | 98.629 |

**L209-213:** Equation (10):

- What is the meaning of the vector of ones "1". Why the

The vector of ones "1" means the all elements of this vector are equal to one (line 204).

- What is $u_j$? explanation needed.

$u_j$ is the $j^{th}$ bias estimation. It is now clarified (line 207). Thank you.

**L224: Figure 3**

- L588: "during the study period" should simply state the study period, e.g., from *** to **

The study period is specifically included for the caption of Figure 3.

**L234:** "The increased LSW just above the boundary layers could be caused by common inversion layers in the troposphere of some oceans."

As discussed in the major comment earlier, there is not any direct definition and calculation of PBL height is this study. In addition, regions especially over tropics might not have clearly

defined PBL due to convection. Would be more appropriate to simply using the height (e.g., 2 -3 km).

Thank you for your suggestion. We now remove the use of PBL and provide the specific height instead (line 226).

- Lager LSW due to "inversion layers", is this a pure guess or with certain supportive evidence? If so, the reference needs to be listed.
This is mentioned in Sokolovskiy et al. (2014). We now provide the reference to support this statement (line 227).
Sokolovskiy, S., W. Schreiner, Z. Zeng, D. Hunt, Y.-C. Lin, and Y.-H. Kuo (2014), Observation, analysis, and modeling of deep radio occultation signals: Effects of tropospheric ducts and interfering signals, Radio Sci., 49, 954–970, doi:10.1002/ 2014RS005436.

L244: "The LSW over ocean below 4 km increases faster over the ocean, and the second peaks value at the PBL top is much larger"

- Again, the "PBL top" should be replaced with 2 km.
We have replaced "PBL top" to a specific height (2km) (line 237).

L254: "thermodynamic structure in the PBL" → "thermodynamic structure in the lower troposphere.
Corrected (line 244). Thank you.

L267: "temperature and specific humidity averaged below 1.5 km"

- Discussion on the sensitive of the result due to the penetration issue should be added.
- We note that taking only the RO profiles penetrating 0.5 km will modify the characteristics of Figure 6 in two aspects. First, the REFB below 10 km becomes positive for LSW/2 larger than 28%. Second, the REFB for cold temperature shows negative at 15 km. The former feature is related to the early cutoff height in the tropical occultation over central Africa. The latter feature is attributed to the inversion associated with the large-scale subsidence near the tropopause near mid-latitude (line 263-268). Sensitivity tests with respect to sampling issues will be discussed in subsection 4.2.

L270: "Instead, N-REFB appears at the PBL top"

- Again, please not use PBL top, but using the height, e.g., ~ 2km
We have replaced "PBL top" to a specific height (2km).

L286: Figure 7

- It is not clear why ONLY the relationship between REFB and LSW over the southern hemisphere (austral summer) were discussed.

We focused on the relationship between REFB and LSW over the southern hemisphere to emphasize the warm and moist conditions during austral summer.
To address the reviewer's comment, we now include the same plot over the northern hemisphere.

- The relationship over norther hemisphere should also be discussed. Same for Figure. 8

Following the reviewer's suggestion, we now revise Figures 7 and 8 to show the relationship between REFB and predictors over the northern hemisphere.

[Figure]

**Figure D: Relationship between LSW/2 and REFB. The solid and dashed lines represent the REFB computed from the statistical model for the ocean and land, respectively, as a function of LSW/2 (Southern Hemisphere only). LSW/2 and REFB are averaged below 1.5 km.**

As shown in Fig. D-b, there is still strong relationship over ocean and land, except that the one over land has a somewhat stronger quadratic feature. Therefore, the strong relationship and conclusion remain for the northern hemisphere. We have included the relevant discussion at line 281-288.

- L604: Can't see the orange open circle (Land data over 1.5 km). could consider using aslightly larger open circle to avoid overlapping by the solid circles.

Thank you for your suggestion. This figure is now re-plotted with bigger open circles for better illustration.

  o Figure caption: "N-biases" should be replaced with REFB to be consistent with manuscript.

  Thank you! We have replaced "N-biases" with REFB to ensure the consistency throughout the manuscript.

- What about the relationship in the northern hemisphere?
  - With the weaker N-bias during the evaluation period, will the relationship holdup? If not, please explain.

    Although the REFB over the northern hemisphere is weaker during the study period, the relationship between REFB and LSW still holds well.
  - Note there are seasonal variations in N-bias, the northern hemisphere is expected to see higher N-biases in NH summer (JJA) season, will the authorsexpect to see clearer relationship in the NH as compare to the Fig. 7?

    Given the strong relationship between REFB and LSW over northern and southern hemisphere, we expect that the relationship during the NH summer season will hold. This is also addressed in line 286-287.

L293: N-biases → REFB

Corrected. Thank you.

L294: Figure 8

- L297: The manuscript only discussed Fig. 8a, but not the Fig. 8b. Discussion of both are required, and the difference should be clearly stated.
- L609: No labels: (a) (b) in Fig. 8. Also Figure caption should be self-explanatory. The manuscript also did not clearly tell the difference. Caption should clearly indicate the difference between the two panels, one for ocean and the other over land, right?
- The relationship is hard to be seen clearly.
- Similar to Fig. 7, the relationship over norther hemisphere should also be presented and discussed.

We apologize for the unclear description of the figure caption in the previous version. This figure is replotted with respect to variable $Q$ and $TQ$ for better illustration. The new Figure 8 shows a clear linear relationship between REFB, $Q$ and $TQ$.

Given a fixed $Q$, the cooler temperature (i.e. smaller $TQ$) will lead to larger negative REFB.

We also expanded our discussion for Fig. 8b to emphasize the difference between the regression model over ocean and land.

For the simulated REFB in Southern Hemisphere, the regression model shows a better fitting performance over ocean than over land (0.79 vs. 0.72) with a stronger quadratic relationship with the $Q$ variable. We also include figures over Northern hemisphere in Fig. 8c and 8d.  Results show that the relationship becomes more linear, i.e. dependence to $Q^2$ is smaller in Northern Hemisphere.

Please see the discussion about Figure 8 in the revised paragraph (line 289-300).

L318: rewrite "inversion layers on top of the surface cold atmosphere"? Do you mean on top of cool sea surface?

Yes! This sentence (line 319) is corrected as "on top of the cool sea surface".

L351: Figure 10

- "Almost all the large RMSEs in the LSW or TQ estimation are removed by the MVEmethod (Fig. 10c and 10f)"

  o It is hard to see the significant improvement from MVE results (10 e,f) as compared to TQ-based approach (10 c,d). Actually, a slight degradation can be seen.

    We are sorry that the differences between Fig. 10c and 10e (and Fig. 10d and 10f) are difficult to tell with the light color we chose. To address this comment, we change the color applied on this figure (the Figure 11 in the revised manuscript). Over the ocean, the improvement with the MVE method over the LSW or TW estimation is evident (with darker blue). However, as the reviewer pointed out, there is slight degradation over the continents of south America and middle Africa.

    We now modify the sentence (line 347-350) to "*The large RMSEs in the LSW or TQ estimation over the oceans are largely removed by the MVE method. However, slight degradation is observed over the continents of south America and middle Africa.*"

  o Need to explain why the training data set show the large improvement of T/Q over ducting region is SE Pacific/Atlantic in Fig. 10-c,e (e.g., SE Pacific [30S,90W]),but was not the case for the testing data set (Fig. 10-d,f)

    We greatly appreciate this insightful comment. From the new color scale used in Figure 11, it is clear that there is little improvement with the MVE method over the TQ estimation in the ducting regions over SE Pacific/Atlantic, as the reviewer pointed out. By examining the T and Q profiles of the training and testing data in the area of 85W to 75W and 30S to 43S, we found that the near-surface temperature and moisture in the testing data are higher than those in the training data (Fig. E). In particular, the moisture difference is as large as 3g/Kg near the

surface. Such high moist condition results in a larger negative REFB with the TQ estimator. The overestimation of the testing data in this region suggests that a statistical model applicable to a broader range of temperature and moisture requires more data.

We have included the discussion in line 355-359.

[Figure]

**Figure E Vertical profiles of the temperature and specific humidify averaged in the area of (85°W to 75°W, 43°S to 30°S).**

Furthermore, with the reviewer's suggestion to separate the data below 0.5km and between 0.5km and 1.5km, we notice that the TQ-based REFB tends to overestimate the negative bias below 0.5 km in the ducting regions. When limiting the RO data between 0.5 and 1.5 km, the TQ estimation well represent the large negative REFB in the ducting regions.

L366:  EQ < Lat < 10°N → 0° < Lat <10°N

Corrected. Thank you.

L383:  3 N → 3 N-unit

Corrected. Thank you.

L386: Please remove the PBL in "positive bias above the PBL altitudes of 3 to 5 km."

Corrected. Thank you.

L423: "… temperature and moisture from the ERA5 reanalysis may have their own bias

Corrected. Thank you.

L612: Box A & B should include the [lat, lon] information.

The [lat, lon] information for Boxes A to C are included in the caption of Figure 14. Thank you.

---

## Referee Report (RR1)

Review of revised paper by Pham et. al. *Estimating the refractivity bias of Formosat-7/COSMIC-II GNSS Radio Occultation in the deep troposphere.*
Richard Anthes
14 February 2024

The authors have responded carefully to my review and except for some minor edits and clarifications, the paper is acceptable for publication.

Comments on revised paper

1. The response to my comment 8 (shown below) was not clear. There are two versions of the 1D-Var retrievals of temperature and water vapor from RO data. The original one, developed more than 20 years ago, is designated wetPrf. The newer, and improved version (Wee et al. 2022) is designated wetPf2. I think the authors used the original version, wetPrf, but it is not clear from their response because they use the term wetPrf2, which does not exist (I know, it can be confusing!). In the revised paper (lines 140-141) they say they use wetPf2, but they reference Wee (2018). If they use wetPf2, they should reference Wee et al. (2022). Please say clearly in the revised version whether they used the original version or the new version.

My original comment and the response:
*8. Line 194—what 1D-Var algorithm was used? The original CDAAC wetPrf or the new CDAAC wetPf2 (Wee et al. 2023)? Or some other one?*
*The wet product we used in the study is the 1D-var wetPrf2. The wetPf2 refractivity is derived from the*
*bending angle profile of atmPrf and through a variational regularization of Abel transform (Wee, 2018),*
*instead of the traditional Abel inversion. We have clarified that the T and Q obtained from the 1D-Var analysis of the RO wet products in line180.*

2. The Rocken et al. (1997) paper was inadvertently omitted in the text of the revised paper. The negative N bias was first noted by Rocken et al. (1997), although the reasons for it were not discovered until a few years later. The reference should be mentioned in the text. Perhaps in line 60: "It is known since 1997 that negative biases in refractivity exist in the lower troposphere, especially in the tropics (Rocken et al. 1997)"
3. The recent paper on detection of superrefraction by Sokolovskiy et al. (2024) should be added to the references in line 53.
4. Line 26—delete "tend to"
5. Line 56-replace "are" with "is" and delete "and regression coefficients"
6. Line 68-replace "choose" with "choosing"
7. Lines 297-298-"drier conditions"
8. Line 362-"investigates"
9. I don't understand the sentence in lines 461-462. Factors such as temporal variations, topography and meteorological effects are implicitly considered in this study, which is

based on statistics of data that include these factors. Possible biases in the ERA5 analysis is one limitation. Fewer data in high latitudes is another. A third is that the statisical results are not perfect, as indicated in the RMSE and other parts of the paper (e.g. Fig. 10). So the results may be useful in a statistical sense and using the polynomials to determine the negative N bias for individual profiles may have errors.

10. Line 471-I think there was only one anonymous reviewer.

Reference
Sokolovskiy, S., Z. Zeng, D. Hunt, J.-P. Weiss, J. Braun, W. Schreiner, R. Anthes, Y.-H. Kuo, H. Zhang, D. Lenschow, and T. VanHove, 2024: Detection of super-refraction at the top of the atmospheric boundary layer from COSMIC-2 radio occultations. *J. Atmos. and Ocean Tech.*, **40**, 65-78. https://doi.org/10.1175/JTECH-D-22-0100.1

---

## Author Response (AR2)

Dear Dr. Anthes,

We deeply appreciate your careful review again for improving our manuscript. We also thank you for pointing out the confusing description about the retrieval product we used in this study. The product we used is wetPf2. As you pointed out, the reference should be Wee et al. (2022). Please see our point-to-point responses to your comments as follows.

1. The response to my comment 8 (shown below) was not clear. There are two versions of the 1D-Var retrievals of temperature and water vapor from RO data. The original one, developed more than 20 years ago, is designated wetPrf. The newer, and improved version (Wee et al. 2022) is designated wetPf2. I think the authors used the original version, wetPrf, but it is not clear from their response because they use the term wetPrf2, which does not exist (I know, it can be confusing!). In the revised paper (lines 140-141) they say they use wetPf2, but they reference Wee (2018). If they use wetPf2, they should reference Wee et al. (2022). Please say clearly in the revised version whether they used the original version or the new version.

    In our paper, we used the new version of 1D-Var retrievals of temperature and water vapor (wetPf2). The data was obtained from Taiwan Analysis Center for COSMIC (TACC) (https://tacc.cwa.gov.tw/data-service/fs7rt_tdpc/level2/). We modified the product name and the reference (Wee et al. 2022). Please see the modifications at line 144 and 184.

2. The Rocken et al. (1997) paper was inadvertently omitted in the text of the revised paper. The negative N bias was first noted by Rocken et al. (1997), although the reasons for it were not discovered until a few years later. The reference should be mentioned in the text. Perhaps in line 60: "It is known since 1997 that negative biases in refractivity exist in the lower troposphere, especially in the tropics (Rocken et al. 1997)"

    Thank you very much for pointing out this important reference. We apoloze for our inadvertent mistake. This sentence is now included accordingly.

3. The recent paper on detection of superrefraction by Sokolovskiy et al. (2024) should be added to the references in line 53.

    Thank you for your suggestion. Sokolovskiy et al. (2024) is now cited at line 54 and included in the reference.

4. Line 26—delete "tend to"

    This sentence is modified accordingly.

5. Line 156-replace "are" with "is" and delete "and regression coefficients"

This sentence is modified accordingly.

6. Line 168-replace "choose" with "choosing" s

The grammar error is corrected. Thank you.

7. Lines 297-298-"drier conditions"

Thank you. It is corrected.

8. Line 362-"investigates"

Thank you. The grammar error is corrected.

9. I don't understand the sentence in lines 461-462. Factors such as temporal variations, topography and meteorological effects are implicitly considered in this study, which is based on statistics of data that include these factors. Possible biases in the ERA5 analysis is one limitation. Fewer data in high latitudes is another. A third is that the statisical results are not perfect, as indicated in the RMSE and other parts of the paper (e.g. Fig. 10). So the results may be useful in a statistical sense and using the polynomials to determine the negative N bias for individual profiles may have errors.

Thank you for your suggestion. We removed this sentence but emphasized that predictors used in the statistical models may not be perfect to capture all attributions of REFB (line 468).

10.    Line 471-I think there was only one anonymous reviewer.

Thank you for pointing this out. It is corrected.